# Transformation-induced stress at telomeres is counteracted through changes in the telomeric proteome including SAMHD1

Jana Majerska[1,2], Marianna Feretzaki[1,2], Galina Glousker[1,2], Joachim Lingner[1,2]

**Telomeres play crucial roles during tumorigenesis, inducing cellular senescence upon telomere shortening and extensive chromosome instability during telomere crisis. However, it has not been investigated if and how cellular transformation and oncogenic stress alter telomeric chromatin composition and function. Here, we transform human fibroblasts by consecutive transduction with vectors expressing hTERT, the SV40 early region, and activated H-RasV12. Pairwise comparisons of the telomeric proteome during different stages of transformation reveal up-regulation of proteins involved in chromatin remodeling, DNA repair, and replication at chromosome ends. Depletion of several of these proteins induces telomere fragility, indicating their roles in replication of telomeric DNA. Depletion of SAMHD1, which has reported roles in DNA resection and homology-directed repair, leads to telomere breakage events in cells deprived of the shelterin component TRF1. Thus, our analysis identifies factors, which accumulate at telomeres during cellular transformation to promote telomere replication and repair, resisting oncogene-borne telomere replication stress.**

## Introduction

Telomeres play critical roles in the progression of human cancer (Maciejowski & de Lange, 2017). Most somatic cells in the human body do not express telomerase (Kim et al, 1994). Therefore, telomeres shorten with every round of DNA replication because of the end replication problem and the nucleolytic processing of chromosome ends (Soudet et al, 2014) by approximately 50–100 bp. Upon reaching a critically short length, telomeres elicit a DNA damage response (DDR) involving the DNA checkpoint protein kinases ATM and ATR (d'Adda di Fagagna et al, 2003; Denchi & de Lange, 2007). The telomeric DDR induces permanent cell cycle arrest referred to as cellular senescence with a G1 DNA content. This block

to proliferation of precancerous cells can be prevented through inactivation of the p53 and RB tumor suppressors (Shay & Wright, 2005). Cells that bypass cellular senescence will hit telomere crisis during which telomeres lose their protective roles from end-to-end chromosome fusions by classical and alternative nonhomologous end joining (Jones et al, 2014). Thus, telomere crisis leads to chromosome fusions, mitotic missegregation, and chromosome breakage events that give rise to extensive chromosome instability. In cancer, telomere crisis is mostly overcome through up-regulation of the telomerase catalytic subunit hTERT, which frequently involves mutations in the *hTERT* promoter (Horn et al, 2013; Huang et al, 2013). Thus, telomerase becomes active, stabilizing telomere length of partially rearranged chromosomes.

In addition to gradual telomere shortening induced by the lack of telomerase, telomeres can be damaged and lost because of stochastic replication defects occurring during semiconservative replication of telomeric DNA (Miller et al, 2006; Chang et al, 2007; Sfeir et al, 2009). Telomere replication defects can give rise to a fragile phenotype, which is characterized by discontinuities in the telomeric signal detected on metaphase chromosome spreads (Sfeir et al, 2009). Telomeres are difficult to replicate and fragile for at least four reasons. First, the single-stranded TTAGGG repeat (G-rich)–containing strand may adopt highly stable G-quadruplex structures that need to be unwound to serve as a template during replication (Sfeir et al, 2009; Paeschke et al, 2011; Vannier et al, 2012). Second, telomeres can fold into t-loop structures in which the telomeric 3′ overhang is tucked into the double-stranded part of the telomere which need to be unwound during replication (Vannier et al, 2012; Doksani et al, 2013). Third, telomeres are transcribed into the long noncoding RNA TERRA that can form DNA/RNA hybrid structures and as such can interfere with replication (Balk et al, 2013; Pfeiffer et al, 2013; Sagie et al, 2017). Fourth, telomere replication is driven from origins of replication that are present in the subtelomeric DNA. Origin firing occurs only rarely from within telomeric repeat sequences (Drosopoulos et al, 2015). Therefore, telomere replication is unidirectional and stalled forks may not be rescued from converging forks coming from the end of

[1]School of Life Sciences, Ecole Polytechnique Fédérale de Lausanne, Lausanne, Switzerland   [2]Swiss Institute for Experimental Cancer Research, Ecole Polytechnique Fédérale de Lausanne, Lausanne, Switzerland

Correspondence: joachim.lingner@epfl.ch

the chromosome. Telomere fragility may become particularly pronounced during oncogenic transformation. Oncogene-induced hyperproliferation leads to replication stress, causing DNA damage at telomeres and elsewhere in the genome leading to cellular senescence (Bartkova et al, 2006; Di Micco et al, 2006; Suram et al, 2012). Remarkably, oncogene-induced damage at telomeres appears to persist, enforcing oncogene-induced cellular senescence (Fumagalli et al, 2012).

Despite the extensive knowledge on the critical roles of telomere length for cellular senescence and telomere crisis during carcinogenesis, very little is known about changes in telomeric protein components that occur during transformation. The telomerase catalytic subunit hTERT, which associates with telomeres during extension, is a notable exception having well-described key roles to allow cell immortality of cancer cells (Bodnar et al, 1998). In addition, mutations affecting the single-strand telomeric DNA-binding protein POT1 have been reported in a range of malignancies (Quesada et al, 2013; Ramsay et al, 2013; Robles-Espinoza et al, 2014; Calvete et al, 2015). However, a systematic analysis of the telomeric proteome during transformation is missing.

Here, we exploit the established in vitro transformation protocol of Hahn and Weinberg (Hahn et al, 1999, 2002) for oncogenic transformation of human lung fibroblasts (HLFs) by introducing defined genetic elements. We apply the previously developed quantitative telomeric chromatin isolation protocol (QTIP) (Grolimund et al, 2013; Majerska et al, 2017) to compare the telomeric proteomes during different stages of transformation. The telomeric proteome changes most notably upon transduction with the SV40 early region expressing the large T and small t antigens inhibiting p53 and the protein phosphatase 2A, respectively. Up-regulated telomeric proteins include factors that safeguard telomeres from fragility and, therefore, may suppress oncogene-induced replication stress and senescence. We also discover a crucial function for SAM domain and HD domain-containing protein 1 (SAMHD1), which counteracts telomere breakage events in cells depleted of telomere repeat-binding factor 1 (TRF1).

## Results

### Experimental system to study the telomeric proteome during transformation

We transformed HLFs (Ducrest et al, 2001) using the following consecutive steps. First, HLFs were transduced with a retroviral vector containing the hTERT cDNA. Thus, HLFs expressed active telomerase, stabilized their telomere length around 13 kb (Fig S1), and became immortal. These cells are referred to as HLF-T. Second, we transduced the HLF-T with retrovirus expressing the SV40 early region, encoding large T and small t antigens. These cells, referred to as HLF-TS suppressed p53 and RB function via large T and phosphatase 2A via small t antigens (Hahn et al, 2002). In a final step, HLF-TS cells were transduced with retrovirus containing the H-RasV12 allele, which is a constitutively active version of the H-Ras GTPase, stimulating uncontrolled cell growth in the absence of external signals. These cells were referred to as HLF-TSR. Upon large-scale cell expansion, HLF-T, HLF-TS, and HLF-TSR were indistinguishable with regard to their telomere length, whereas the

telomeres in HLFs with inactive telomerase were markedly shorter (Fig S1), because of progressive telomere attrition (Cristofari & Lingner, 2006).

We compared the telomeric protein composition of the four HLF cell lines by QTIP (Grolimund et al, 2013; Majerska et al, 2017) in a pairwise fashion (Fig 1A). Briefly, cells to be compared were grown in different SILAC (stable isotope labeling by amino acids in cell culture; Ong et al, 2002) media to distinguish identical proteins from the pair of cells by their mass and be able to identify quantitative differences in telomere protein abundance. Cell populations to be compared were mixed (Fig 1B) and cross-linked with formaldehyde and ethylene glycol bis(succinimidyl succinate). Chromatin was sonicated and telomeric chromatin isolated using antibodies against the abundant telomeric proteins TRF1 and TRF2. Protein amounts were compared by LC-MS/MS. Replicates were done in which the heavy and light SILAC media were swapped between the two cell populations (Fig S2). The recovery of telomeric DNA was approximately 5–10% and the enrichment of telomeric DNA over Alu repeat DNA around 300-fold (Figs 1C and S3). The spectral counts of shelterin proteins obtained with TRF1/TRF2 antibodies varied from roughly 50 for POT1 and TPP1 to more than 500 for TRF2 (Fig 1D). Shelterin proteins were completely absent from control IgG pulldowns, confirming highest specificity of the purification. Proteins were considered as putatively telomeric when their spectral counts obtained with TRF1/2 antibodies over those obtained with IgG were significantly higher in at least two of the four QTIPs using a Significance A left-sided test (P < 0.05). 134 proteins passed the filtration criteria (Table S1). The identified proteins overlapped partially with proteins that were identified in previous telomeric proteomic screens (Dejardin & Kingston, 2009; Grolimund et al, 2013; Bartocci et al, 2014). These include most prominently the six shelterin proteins but also proteins involved in DNA replication (MCM4, MCM5, and MCM7), repair and end processing (RAD50; Apollo), chromatin remodeling (SMCHD1), and RNA processing (TCEA1) (Fig 2A). In addition, several new telomeric protein candidates were discovered in HLFs. To test their association with telomeric DNA, we expressed selected candidates (Fig 2B) as 3xHA-tagged proteins upon transient transfection in HEK293T cells and tested association with telomeric DNA and Alu repeat DNA by chromatin immunoprecipitation (ChIP) via the HA tags (Figs 2C and S4). HA-tagged CLP1, PNUTS, SAMHD1, ARHGAP1, and TCEA1 all immunoprecipitated with the anti-HA antibody significantly more telomeric DNA than the empty vector control, indicating that they are genuine telomere-associated proteins. The considerably lower but detectable association with Alu repeat DNA suggested association also with other regions of the genome.

### Identification of telomeric proteins that change during transformation

The four comparisons by QTIP (Fig 1B) identified 39 telomeric factors that became either up- or down-regulated during different steps of transformation (Figs 3 and S5, and Table 1). The most notable differences induced during transformation could be ascribed to expression of the SV40 early region. Several proteins involved in DNA replication, DDR, and DNA repair became up-regulated at telomeres upon expression of the T antigens. H-RasV12 expression

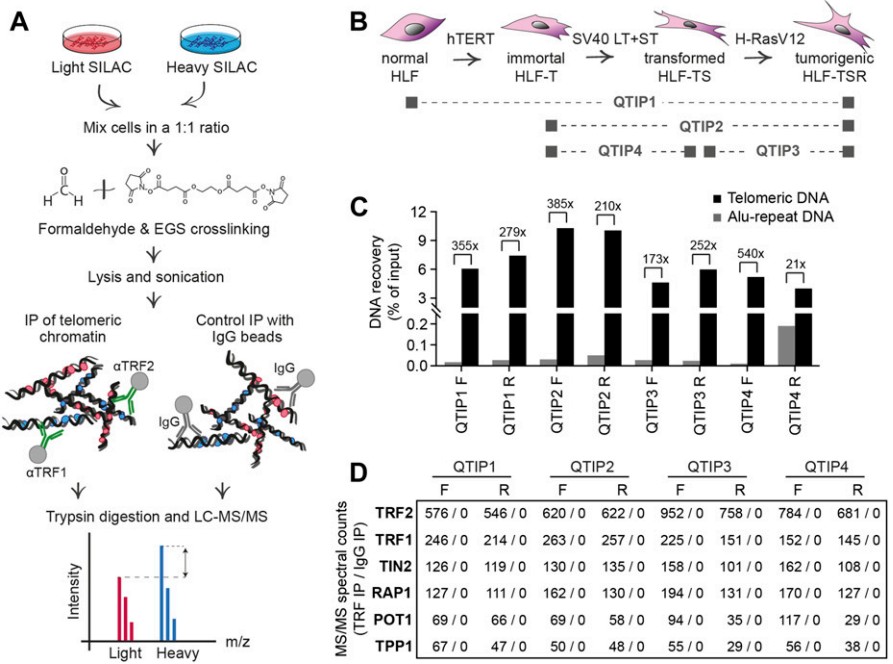

**Figure 1. QTIP method was used to characterize changes in telomere protein composition during cellular transformation.**

**(A)** Workflow of QTIP. **(B)** Schematic of the cell line model and overview of the four pairwise QTIP experiments. HLFs were serially transduced with retroviral vectors expressing hTERT, the SV40 large T (LT) and small t (ST) antigens, and H-RasV12. **(C)** Quantification of precipitated DNA in QTIPs, based on a dot blot hybridized with a specific telomeric probe and a control Alu repeat probe (Fig S3). To determine IP efficiency, the amounts of telomeric DNA in QTIP eluates were quantified and compared with the telomeric DNA in inputs. Fold enrichment of precipitated telomeric DNA compared with precipitated Alu repeat DNA is used as an indicator of IP specificity. Plotted are values from the forward (F) and reverse (R) TRF IP replicates. **(D)** Enrichment of shelterin subunits in QTIPs. The mean of spectral counts of two replicates is indicated.

was associated with milder alterations and contributed to down-regulation of a small subset of proteins of as-yet-unknown telomeric function. In some cases, H-RasV12 slightly counteracted the effect of SV40 T antigens (Fig 3B). The transformation-induced changes did not depend on telomere length because the comparison of HLF-TSR with either HLF or HLF-T cells yielded similar results. Cell transformation also increased the levels of shelterin components TRF1, TRF2, and its interacting protein RAP1, whereas TIN2, TPP1, and POT1 did not change (Figs 3 and S5). Up-regulation of TRF2 was confirmed on Western blots (Fig S6A) and is consistent with previous reports of increased TRF1

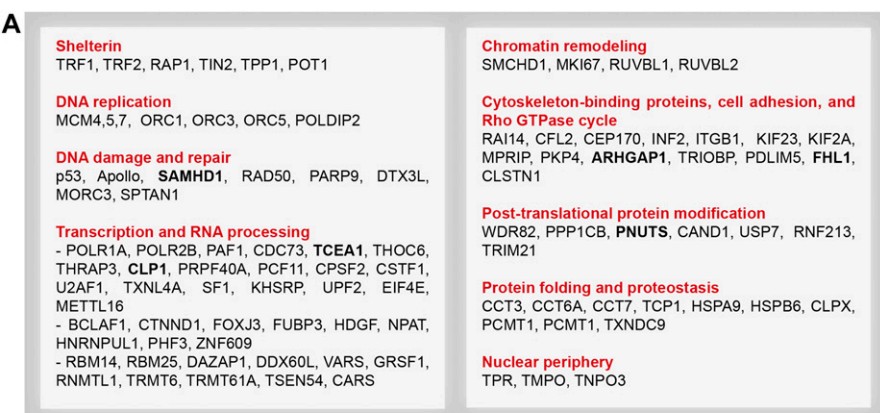

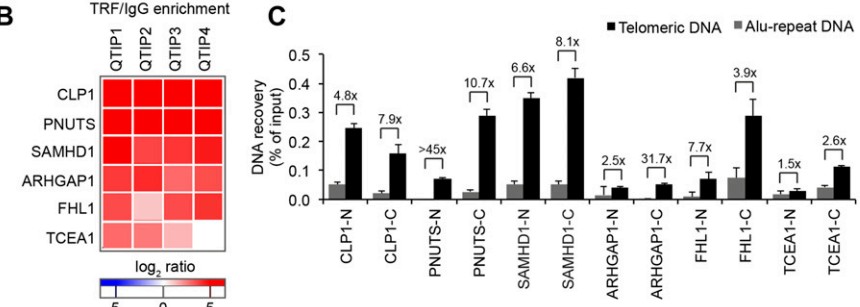

**Figure 2. Telomeric proteome in human fibroblasts.**
**(A)** Overview of proteins with significant TRF/IgG enrichment in at least two of four QTIPs (Significance A left-sided test, $P < 0.05$). The full protein list is in Table S1, list B. **(B, C)** Validation of telomeric localization of selected QTIP candidates by anti-HA ChIP against N- and C-terminally tagged proteins. The precipitated DNA was analyzed as described for Fig 1C. The background of empty vector control was subtracted. Plotted is mean + SD from two to three technical replicates. The images of dot blots are provided in Fig S4.

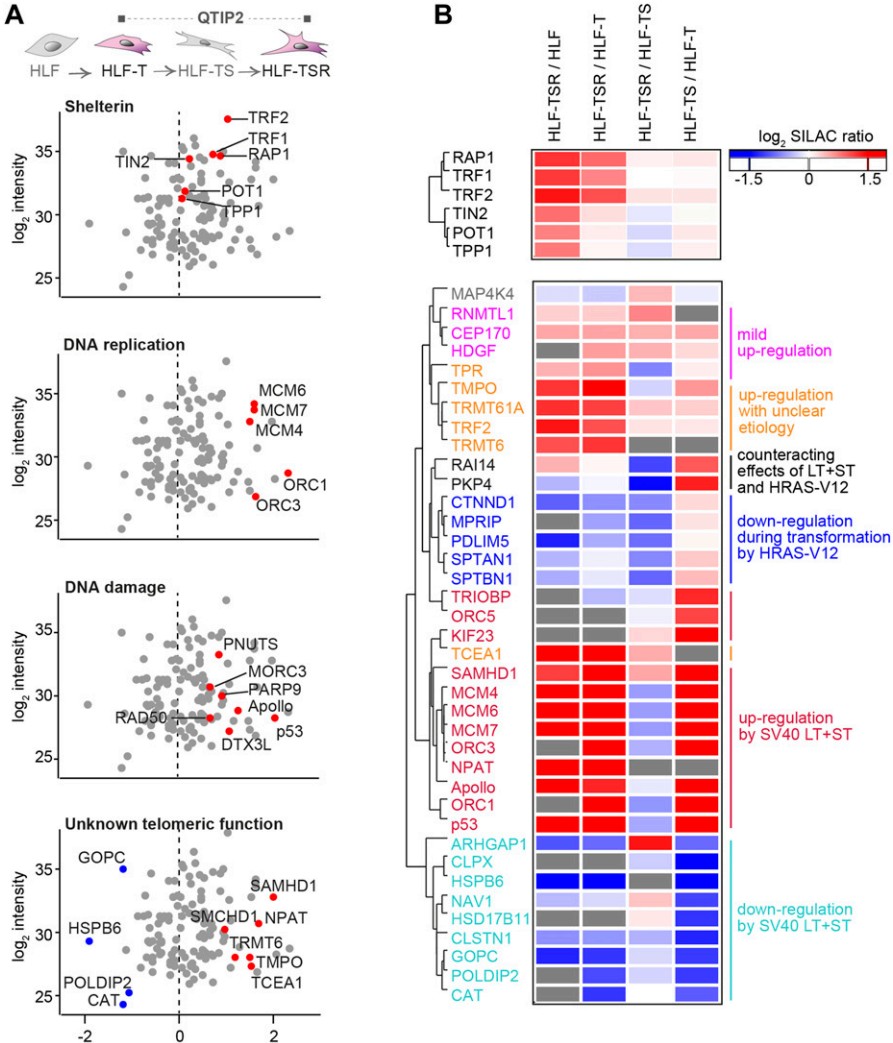

**Figure 3. Identification of transformation-responsive telomeric proteins.**
**(A)** Scatter plots for QTIP2 showing differences in telomere recruitment between HLF-T and HLF-TSR cell lines. Plotted are mean values from the forward and reverse TRF IP replicates. See also Fig S5. **(B)** Heatmap of differentially regulated telomeric proteins. Top panel: shelterin proteins; bottom panel: putative telomeric proteins that were significantly up-/down-regulated in TRF IP in at least one QTIP (Significance B, both sides, $P < 0.05$). Missing values are displayed in gray.

and TRF2 expression in human malignancies (Nakanishi et al, 2003; Diehl et al, 2011; Pal et al, 2015; Chen et al, 2017). Overall, our analysis indicated that factors that preserve telomere integrity became up-regulated during transformation. To further corroborate this notion, we analyzed the proteome of chromatin extracts of the four different HLF lines (Table S1). This analysis confirmed that factors which contribute to genome stability are up-regulated during transformation (Fig S6B), whereas the increase of phosphorylated ATM pS1981 and Chk2 pT68 indicated increased DNA damage (Fig S6A). The up-regulation of p53 with SV40 large T expression was expected as SV40 large T inactivates p53, preventing the up-regulation of the p53 target MDM2, which mediates p53 degradation in a negative feedback loop. Consistently, the p53 target p21 decreased upon transduction with SV40 large T, suggesting efficient suppression of p53 function (Fig S6A). Overall, the analysis suggested that the increased DNA damage at telomeres and elsewhere in the genome, which may result from oncogene-induced replication stress and hyperproliferation (Fig S6C and D), is counteracted through SV40 T antigen-induced up-regulation of proteins that contribute to DNA stability and repair.

## TERRA is up-regulated during cellular transformation

The telomeric long noncoding RNA TERRA has been implicated among others in the telomeric DDR and the regulation of telomerase (Azzalin & Lingner, 2015). In addition, TERRA can form RNA/DNA hybrid structures at telomeres, which represent obstacles for the replication machinery (Balk et al, 2013; Pfeiffer et al, 2013) while promoting homology-directed repair (Arora et al, 2014; Graf et al, 2017). We measured overall TERRA levels on Northern blots (Fig 4A) and TERRA molecules stemming from specific chromosome ends by quantitative RT-PCR (RT–qPCR) (Feretzaki & Lingner, 2017) (Fig 4B). This analysis revealed that the overall TERRA levels increased with every step of transformation. The analysis of individual TERRA molecules showed telomere end-specific regulation. TERRA stemming from telomeres 9p and XpYp increased most notably upon transduction with hTERT, whereas 2p TERRA increased upon transduction with the SV40 early region (Fig 4B). 15q, 17q, and 20q TERRA were not up-regulated during transformation. We also tested if the increased TERRA levels gave

**Table 1. Transformation-responsive telomeric proteins identified by QTIP, reported functions, and association with cancer.**

| Gene name | Major pathway | Canonical function | Cancer association |
|---|---|---|---|
| APOA1 | Lipid metabolism | Participates in cholesterol transport from tissues to liver for excretion; roles in human sperm motility. | (Wang et al, 2009; Su et al, 2010; van Duijnhoven et al, 2011; Jiang et al, 2014; Zamanian-Daryoush & DiDonato, 2015) |
| ARHGAP1 | Rho GTPase signaling | GTPase activator for the Rho, Rac, and Cdc42 proteins. | (Ahn et al, 2012; Li et al, 2017) |
| CAT | Antioxidant activity | Decomposition of hydrogen peroxide. | (Bauer, 2012; Glorieux et al, 2015; Wang et al, 2016) |
| CEP170 | Microtubule dynamics | Role in microtubule organization and cell morphology. | Unknown |
| CLPX | Proteostasis | Component of the mitochondrial unfoldase–peptidase complex. | Unknown (Seo et al, 2016) |
| CLSTN1 | Intracellular transport | Regulates kinesin-mediated cargo transport and organizes microtubule polarity during axon development. | Unknown |
| CTNND1 | Cell adhesion, transcription | Transcription regulation, cell adhesion, Wnt signaling, and spatiotemporal control of small Rho-GTPases. | (Thoreson & Reynolds, 2002; Dann et al, 2014; Kourtidis et al, 2015; Li et al, 2015) |
| DCLRE1B (Apollo) | DNA repair, telomere maintenance | 5'-3' exonuclease, control of DNA damage repair and topological stress, generation and maintenance of telomeric overhangs, and DNA replication. | Unknown (Natrajan et al, 2007; Karami et al, 2016) |
| GOPC | Proteostasis | Intracellular protein trafficking and degradation. | (Charest et al, 2003; Ohara et al, 2017) |
| HDGF | Transcription regulation | Heparin-binding protein with mitogenic activity. Regulates transcription. | (Hu et al, 2003; Chen et al, 2015; Lian et al, 2015; Wu et al, 2016; Yang et al, 2016) |
| HSD17B11 | Lipid metabolism | Converts 5α-androstan-3α, 17β-diol to androsterone. | (Nakamura et al, 2009) |
| HSPB6 | Proteostasis | Acts as a molecular chaperone. | (Noda et al, 2007; Matsushima-Nishiwaki et al, 2011, 2016; Nagasawa et al, 2014; Qiao et al, 2014; Ju et al, 2015) |
| KIF23 | Cytokinesis | Component of the centralspindlin complex, essential for cytokinesis in Rho-mediated signaling. | (Takahashi et al, 2012; Sun et al, 2015, 2016; Kato et al, 2016) |
| MAP4K4 | Protein kinase | Activates several mitogen-activated protein kinase pathways; involved in cancer cell growth, apoptosis, and migration. | (Qiu et al, 2012; Haas et al, 2013; Feng et al, 2016; Liu et al, 2016; Gao et al, 2017) |
| MCM4, 6, 7 | DNA replication | Components of the MCM2–7 DNA replicative helicase. | (Honeycutt et al, 2006; Shima et al, 2007; Bagley et al, 2012; Das et al, 2013; Kwok et al, 2015) |
| MPRIP | Cytoskeleton regulation | Targets myosin phosphatase to the actin cytoskeleton. Regulation of the actin cytoskeleton by RhoA and ROCK1. | (Ono et al, 2008) |
| NAV1 | Cytoskeleton regulation | Regulation of the microtubule cytoskeleton, important for neuronal development, and interacts with the RhoGEF TRIO. | Unknown |
| NPAT | Transcription, cell cycle | Required for progression through the G1 and S phases, for S phase entry, and for activation of histone gene transcription. | (Saarinen et al, 2011; Hamdi et al, 2017) |
| ORC1, 3, 5 | DNA replication | Components of the origin recognition complex; required for the assembly of pre-RC. | Unknown (Champeris Tsaniras et al, 2014) |
| PDLIM5 | Calcium signaling | Regulates intracellular calcium levels by linking calcium channel and PKC. | (Eeles et al, 2009; Kim et al, 2010) |
| PKP4 | Cell adhesion | Regulates RhoA signaling during cytokinesis. | Unknown |
| POLDIP2 | DNA replication and repair | DNA replication and damage repair, also implicated in mitochondrial function, extracellular matrix regulation, cell cycle progression, focal adhesion, and cell migration. | (Grinchuk et al, 2010; Chian et al, 2016) |

**Table 1. Continued**

| Gene name | Major pathway | Canonical function | Cancer association |
|---|---|---|---|
| *RAI14* | Cytoskeleton regulation | May have roles in human testis development and spermatogenesis. | Unknown |
| *RNMTL1* | RNA processing | Catalyses the formation of 2′-o-methylguanosine at position 1370 in the 16S mitochondrial large-subunit rRNA. | (Haiman et al, 2013) |
| *SAMHD1* | dNTP and nucleic acid metabolism | dNTPase activity, has roles in DNA repair, innate immunity, cancer, and HIV-1 restriction; controversial nuclease activity. | (Clifford et al, 2014; Shi et al, 2014; Wang et al, 2014; de Silva et al, 2014; Merati et al, 2015; Rentoft et al, 2016) |
| *SPTAN1* | Regulation of cytoskeleton | A scaffolding protein involved in cell adhesion and motility; implicated in repair of DNA interstrand cross-links. | (Tuominen et al, 1996; Gorman et al, 2007; Cunha et al, 2010; Wolgast et al, 2011; Hinrichsen et al, 2014) |
| *SPTBN1* | Regulation of cytoskeleton | Scaffolding function in protein sorting, cell adhesion, and migration; implicated in TGF-β signaling. | (Gorman et al, 2007; Jiang et al, 2010; Yao et al, 2010; Zhi et al, 2015) |
| *TCEA1* | Transcription | Required for efficient RNA polymerase II transcription elongation past template-encoded arresting sites. | (Hubbard et al, 2008; Shema et al, 2011) |
| *TERF2* | Telomere maintenance | Required for telomere capping and protection. Inhibits nonhomologous end joining and ATM activation at telomeres. Required for t loop formation. | (Nakanishi et al, 2003; Bellon et al, 2006; Munoz et al, 2006; Diehl et al, 2011; Biroccio et al, 2013; Pal et al, 2015; Chen et al, 2017) |
| *TMPO* | Nuclear architecture | Involved in the structural organization of the nucleus and in the post-mitotic nuclear assembly. | (Brachner & Foisner, 2014; Zhang et al, 2016) |
| *TP53* | DDR, cell cycle | Transcription factor inducing cell cycle arrest/senescence and apoptosis; induction of DNA repair genes. | (Muller et al, 2011; Muller & Vousden, 2013; Kaiser & Attardi, 2017) |
| *TPR* | Nucleocytoplasmic transport | Scaffolding element of the nuclear pore complex essential for normal nucleocytoplasmic transport of proteins and mRNAs. | (David-Watine, 2011) |
| *TRIOBP* | Cytoskeleton regulation | Might be involved in actin remodeling, directed cell movement, and cell cycle regulation. | (Bao et al, 2015; Thutkawkorapin et al, 2016) |
| *TRMT6, TRMT61A* | tRNA modification | Catalyses the formation of N1-methyladenine at position 58 in initiator methionyl-tRNA. | (Shi et al, 2015; Macari et al, 2016) |

increased RNA/DNA hybrids at telomeres. The S9.6 monoclonal antibody, which recognizes RNA/DNA hybrids (Hu et al, 2006), was used to perform DNA–RNA immunoprecipitation (Fig S7). The immunoprecipitated nucleic acids contained telomeres, indicating the presence of RNA/DNA hybrid structures consistent with previous analyses (Arora et al, 2014; Sagie et al, 2017). The signal was abolished upon pretreatment with RNase H, which destroys the RNA part in RNA/DNA duplexes, demonstrating the specificity of the assay. However, the amounts of RNA/DNA hybrids at telomeres and at Alu repeats did not change notably during transformation, indicating that increased hybrids at telomeres are not responsible for increased replicative stress that may be induced during transformation.

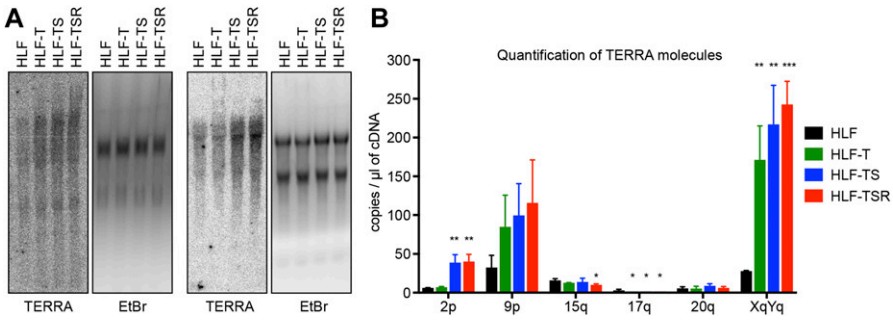

**Figure 4. TERRA levels are elevated during transformation.**
**(A)** Northern blot analysis of total RNA from the HLF-derived cell lines. TERRA was detected using a telomeric DNA probe complementary to the UUAGGG repeats. Ethidium bromide (EtBr) staining is shown as a loading control. Two independent biological replicates are shown. **(B)** TERRA quantification by RT–qPCR with primers specific for the indicated subtelomeric sequences. Plotted is mean + SD from three biological and two technical replicates. Two-tailed unpaired *t* test, comparing all derived cell lines with primary HLFs (*P < 0.05; **P < 0.01; ***P < 0.001; bars lacking asterisks are not significant).

## Up-regulated proteins prevent telomere fragility

To test the hypothesis that during transformation, up-regulated proteins act against replicative stress at telomeres, we depleted candidate proteins (Fig 5A) using siRNA pools in HLF-TSR and HeLa cells. Measuring mRNA levels by RT–qPCR (Fig 5B) revealed efficient depletion of all candidates except for PNUTS, whose depletion strongly impacted on cell viability. Metaphase chromosomes were prepared and telomeric signals were detected by FISH. Replication stress at telomeres gives the so-called fragile telomeres, which are characterized by a smeary telomeric signal or multiple telomeric signals (Fig 5C). Telomere loss, apposition, and breakage ("outside") events were also scored (Table S2). Quantification revealed increases in telomere fragility upon depletion of several candidates. Depletion of NPAT scored only positively in HeLa cells. NPAT has known roles in transcriptional activation of histones but was not known to be also telomeric (Zhao et al, 2000). Its depletion caused accumulation of genome-wide 53BP1 foci, a subset of which colocalized with telomeres (Fig S8). SAMHD1 plays, among others, roles in homology-directed repair (Daddacha et al, 2017). Its roles at telomeres are further characterized and discussed below. DCLRE1B/SNMB1/Apollo is a DNA 5′-3′ exonuclease with roles in telomere end processing (Lenain et al, 2006; van Overbeek & de Lange, 2006; Wu et al, 2012). Apollo also relieves topological stress during telomere replication (Ye et al, 2010). TMPO is a lamina-associated protein with roles in nuclear architecture and thus may contribute to telomere localization (Harris et al, 1995). PARP9 is a poly(ADP-ribose) polymerase involved in DNA repair (Yang et al, 2017). PNUTS binds and regulates protein phosphatase 1, which among others is involved in the telomeric DDR, DNA repair, and DNA replication origin licensing (Kim et al, 2009; Landsverk et al, 2010; Hiraga et al, 2017).

PNUTS depletion caused accumulation of genome-wide 53BP1 foci, a subset of which colocalized with telomeres (Fig S8). SMCHD1 is required for X chromosome inactivation (Nozawa et al, 2013). It preferentially associates with long telomeres (Grolimund et al, 2013) and has reported roles in DNA repair (Coker & Brockdorff, 2014; Tang et al, 2014).

## SAMHD1 counteracts telomere breakage in TRF1-depleted cells

SAMHD1 has been implicated in homology-directed repair of DNA double-strand breaks, recruiting CtIP to promote DNA end resection (Daddacha et al, 2017). In addition, SAMHD1 has been recently shown to stimulate processing of stalled replication forks (Coquel et al, 2018). To assess the roles of SAMHD1 in curbing replication stress or maintenance of telomere intactness, we depleted it in HeLa cells using shRNAs (Fig 6A). As a control, we induced local replication stress at telomeres by depleting TRF1. TRF1 prevents telomere fragility presumably through the recruitment of the BLM and RTEL1 helicases (Sfeir et al, 2009). SAMHD1 depletion enhanced telomere fragility slightly (Figs 5 and 6), whereas TRF1 depletion showed a stronger effect (Fig 6B and C, and Table S3). Strikingly, we observed that the co-depletion of SAMHD1 and TRF1 caused a novel telomere phenotype we referred to as "outsider" in which the telomere signal was detached from the telomeric chromatin of metaphase chromosomes (Fig 6B and D). No notable effects on telomere length could be observed in a TRF analysis in this short-term experiment (Fig S9). Furthermore, the telomeres did not score positive for the presence of extrachromosomal DNA in the form of t-circles, which have been detected in cells that utilize the alternative lengthening of telomeres (ALT) mechanism (Cesare & Griffith, 2004). The most straightforward interpretation of these results is

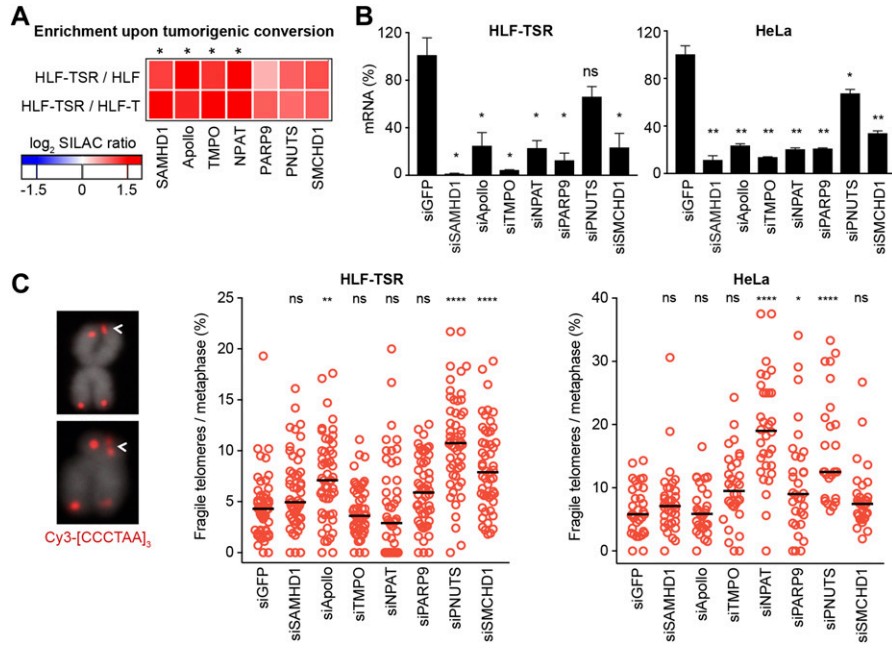

**Figure 5.  Cellular transformation up-regulates proteins that help preserve telomere integrity.**
siRNA screens were performed to test the effect of candidate depletion on telomere structure visualized on metaphase chromosome spreads. **(A)** Heatmap showing behavior of the selected candidates in QTIPs. Reported are mean SILAC ratios from the forward and reverse TRF IP replicates. The asterisks indicate proteins that were classified as significantly "transformation responsive" in the QTIP screen (Table S1, list C). **(B)** RT–qPCR to verify candidate depletion using siRNA pools in HLF-TSR and HeLa cells. Knockdown efficiency was compared with the mRNA levels in cells transfected with control siRNA against GFP. Plotted is mean + SD from two biological and two technical replicates for HLF-TSR and from two technical replicates from one representative experiment for HeLa. Two-tailed unpaired $t$ test (*$P$ < 0.05; **$P$ < 0.01; ns = not significant). **(C)** Analysis of fragile telomeres upon candidate depletion. Indicated telomere aberrations were scored on metaphase spreads with telomeres detected by FISH with a Cy3-[CCCTAA]$_3$ probe (red) and DNA stained with DAPI (gray). HLF-TSR: >50 metaphases from two independent experiments (except for siGFP samples, for which data from three independent replicates) were analyzed for each condition. HeLa: >25 metaphases were analyzed per condition in a single experiment. The black line represents median. One-way ANOVA with Dunnett's multiple comparisons test, comparing all conditions to siGFP (*$P$ < 0.05; **$P$ < 0.01; ****$P$ < 0.0001; ns = not significant). See also Table S2.

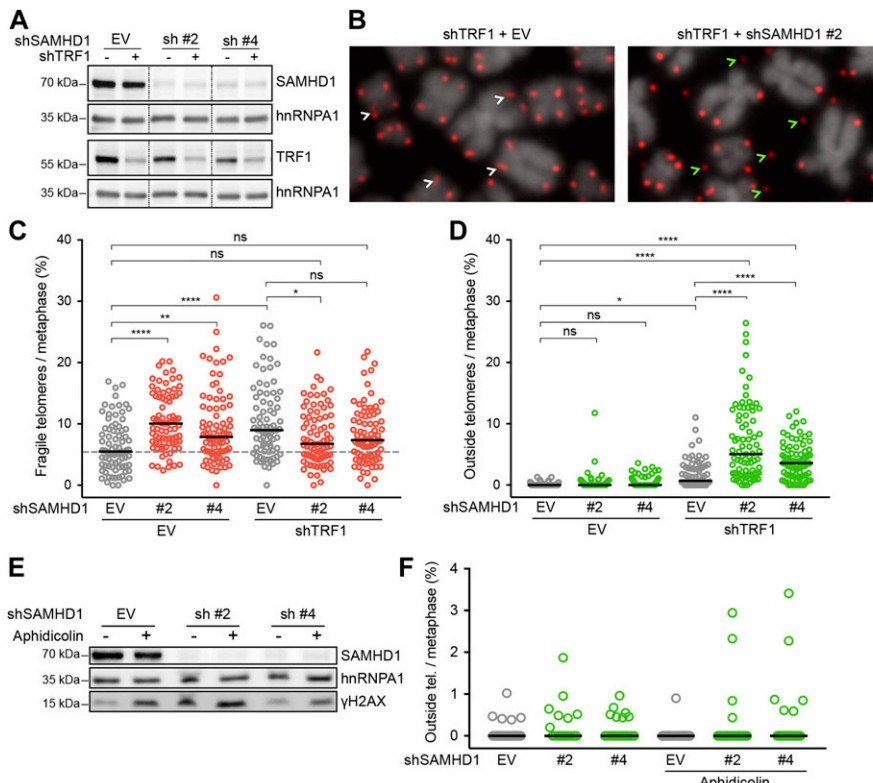

**Figure 6. SAMHD1 is critical for telomere integrity in TRF1-depleted but not in aphidicolin-treated cells.**
**(A–D)** Characterization of telomere aberrations on metaphase chromosomes induced by SAMHD1 knockdown in TRF1-proficient and TRF1-deficient HeLa cells. **(A)** Immunoblotting analysis of knockdown efficiency in cells treated with the indicated shRNAs for 6 d. SAMHD1 and TRF1 are shown on two separate membranes. hnRNPA1 is used as a loading control. Irrelevant lanes have been omitted from the image (dashed lines). **(B)** Representative metaphase chromosome spreads with telomeres detected by FISH with a Cy3-[CCCTAA]$_3$ probe (red) and DNA stained with DAPI (gray). White arrows point to typical fragility (smears and multiple telomeric signals), whereas green arrows indicate outside telomeres (i.e., a telomeric signal positioned outside the DAPI signal). **(C, D)** Quantification of fragile and outside telomeres from cells in (A). 81 metaphases from three independent experiments were analyzed for each condition. The black line represents median. One-way ANOVA with Tukey's multiple comparisons test (*$P < 0.05$; **$P < 0.01$; ****$P < 0.0001$; ns = not significant). **(E, F)** Effect of SAMHD1 depletion on telomere structure in aphidicolin-treated HeLa cells. **(E)** Immunoblot verification of SAMHD1 depletion and DDR induction. Where indicated, the cells were treated with 0.1 µg/ml aphidicolin for 20 h before harvest. **(F)** Quantification of outside telomeric FISH signal on metaphase chromosome spreads from cells in (E). Shown are data from a representative experiment with ≥17 metaphases and >3,000 telomeres analyzed per condition. Differences are not statistically significant. See also Table S3.

that telomeres break in the absence of TRF1 during replication and that SAMHD1 is required for their repair.

We then tested if SAMHD1 depletion also triggers outsider telomeres in cells in which telomere fragility was induced by treatment with the DNA polymerase α and δ inhibitor aphidicolin (Figs 6E and F, and Table S3). However, the outsider phenotype was not triggered in this setting probably reflecting the different defects in TRF1-depleted versus DNA polymerase–inhibited cells.

# Discussion

Changes in telomere length have been known for several decades to promote two key events during tumorigenesis, cellular senescence, and telomere crisis. Here, we discover that cellular transformation also involves crucial changes in telomere protein composition. We identify 134 proteins that are enriched in telomeric chromatin fractions of human fibroblasts. Importantly, 39 of these changed significantly in abundance at telomeres during oncogenic transformation. Many of the up-regulated proteins had been linked to cancer development in previous studies (Table 1); however, the link between their telomere recruitment and tumorigenesis remains unexplored. Most up-regulated telomeric factors play roles in either DNA replication, DNA repair, or telomere protection. The pre-replication complex (pre-RC) components ORC1, 3, and 5 and MCM4, 6, and 7 may be recruited to telomeres via TRF2, which becomes more abundant during transformation. It remains to be tested if the recruited pre-RC components form active origins. Origin firing within

telomeres occurs rarely, but dormant origins within telomeres may be activated upon stalling of replication forks that enter the telomeres from subtelomeric regions (Drosopoulos et al, 2015). We also identify Apollo/DCLRE1B as a transformation up-regulated factor, which has been reported to counteract topological stress at telomeres (Ye et al, 2010). In support of a role in curbing replication stress, Apollo/DCLRE1B depletion increased telomere fragility. In addition, we identify six novel factors that were up-regulated during transformation and whose depletion caused telomere fragility.

Oncogene-induced hyperproliferation leads to replication stress, possibly involving depletion of nucleotides or increased collisions with transcription complexes, although the exact mechanism remains unknown (Bartek et al, 2007; Hills & Diffley, 2014). During transformation of HLFs, TERRA levels but not DNA/RNA hybrids increased. Thus, increased telomere transcription may contribute to telomere fragility, although other mechanisms should not be excluded. We propose that the up-regulation of factors that suppress telomere fragility is critical to resist oncogene-induced replication stress at telomeres during transformation. The fragility-suppressing factors were induced upon expression of SV40 T antigens inactivating p53, RB, and protein phosphatase 2A. Expression of oncogenic H-RasV12, which was introduced as a last step in our transformation protocol, did not contribute to their up-regulation. During natural carcinogenesis, oncogene-induced hyperproliferation precedes inactivation of p53 and RB and therefore oncogene-induced senescence may be promoted (Halazonetis, 2004; Gorgoulis et al, 2005; Bartkova et al, 2006; Di Micco et al, 2006; Halazonetis et al, 2008; Suram et al, 2012). Our results suggest that targeting of p53, RB, and PP2A by SV40 T antigens

not only alleviates checkpoint-induced cell cycle control but also triggers expression of factors that suppress DNA damage upon replication stress, which is prevalent at telomeres (Fig 7A).

One of the top candidates up-regulated upon transformation of HLFs was SAMHD1. Therefore, we decided to explore its functions at telomeres. SAMHD1 is a dNTPase, which restricts HIV-1 replication (Laguette et al, 2011). SAMHD1 is mutated in cancer (Clifford et al, 2014; Merati et al, 2015; Rentoft et al, 2016) and in the neurodegenerative autoimmune disorder Aicardi–Goutières syndrome in which aberrant nucleic acids accumulate, inducing innate immunity (Ballana & Este, 2015). More recently, SAMHD1 functions in homology-directed DNA repair (Daddacha et al, 2017) and DNA replication (Coquel et al, 2018) have been discovered. Specifically, SAMHD1 recruits CtIP to DNA double-strand breaks to facilitate end resection and homologous recombination (HR) (Daddacha et al, 2017). We find that double depletion of TRF1 and SAMHD1 gives outsider

telomeres, which seem broken off from the chromosome arms. We suspect that this phenotype can be ascribed to the functions of SAMHD1 in HR and we propose the following model (Fig 7B). TRF1 depletion gives telomere fragility because of inefficient recruitment of the G-quadruplex unwinding helicases BLM and RTEL1. Therefore, replication forks may stall and are eventually processed into DNA double-strand breaks near the telomere–subtelomere boundary. In SAMHD1-proficient cells, the breaks are repaired by homology-directed repair. In SAMHD1-deficient cells, the broken forks remain unrepaired. However, the telomeric FISH signals of outsider telomeres indicate that telomere replication does proceed downstream of the break, possibly triggered by repriming events or activation of telomeric origins that converge toward the double-strand break (Pasero & Vindigni, 2017). A contribution of telomerase in promoting telomere synthesis downstream of the broken fork cannot be excluded. The latter was seen in fission yeast in which the TRF1-ortholog Taz1 was deleted (Miller et al, 2006). Loss of Taz1 led to

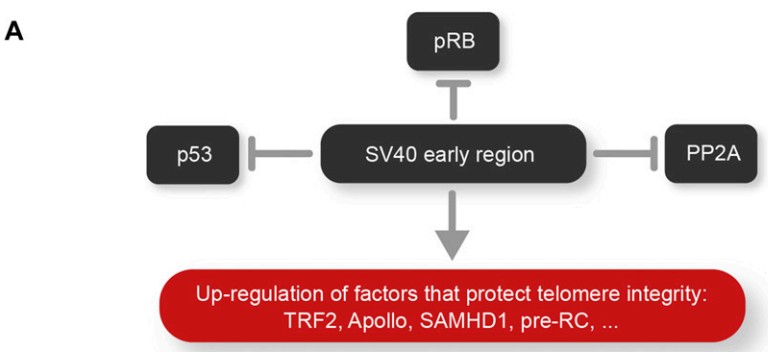

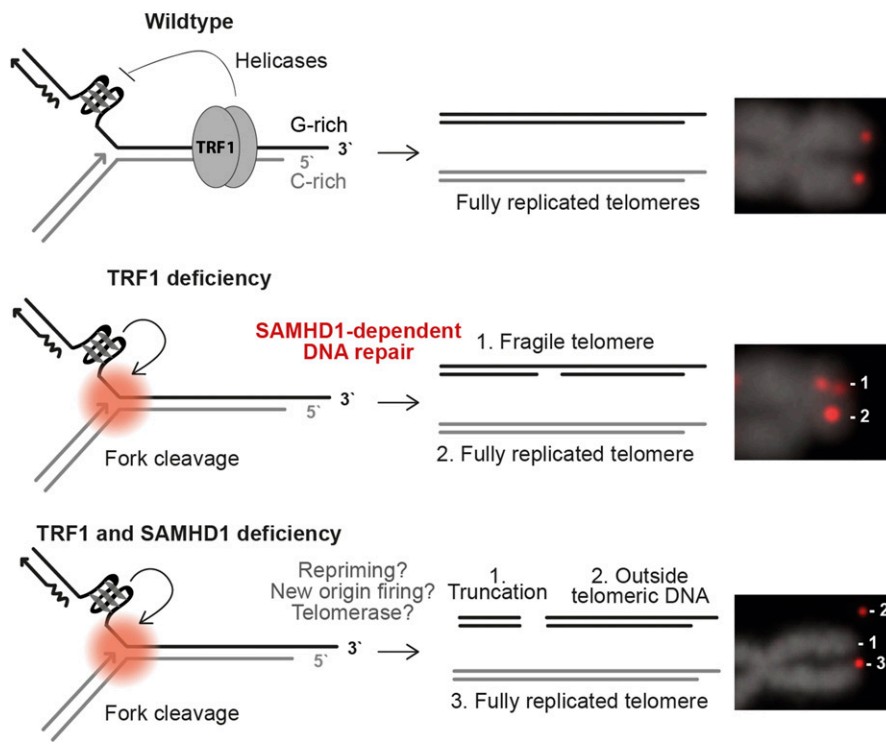

**Figure 7. Hypothetical model for transformation-associated changes at telomeres.**
**(A)** The SV40 large T and small t antigens inhibit p53, RB, and PP2A. In addition, SV40 early gene expression leads to up-regulation of factors, which promote telomere protection and replication. **(B)** Telomere replication requires TRF1. In the absence of TRF1, G-quadruplex structures accumulate at telomeres, which become cleaved during replication and repaired by SAMHD1-dependent repair. In the absence of both TRF1 and SAMHD1, double-strand breaks are retained. Telomere synthesis proceeds downstream of the breaks mediated by replication fork restart, telomerase, or replication fork firing from within telomeric repeats.

stalled replication forks at telomeres, which could be maintained only in the presence of telomerase.

Overall, our study underlines the importance of telomere replication stress in cellular transformation. It identifies a number of novel factors that are induced by SV40 T antigens to counteract telomere fragility. Deregulation of these proteins at telomeres might play an unrecognized role in the development of cancer, as exemplified by SAMHD1, which has been found mutated in cancer cells and seems to have a pivotal function in the preservation of telomere integrity. This work will stimulate further studies to unravel the mechanisms that underlie telomere replication stress and oncogene-induced senescence during transformation.

## Materials and Methods

### Cell lines

The following cell lines were used: HeLa; HEK293T; HLF (Ducrest et al, 2001); HLF-T—HLF cells stably infected with hTERT expressing retroviruses; HLF-TS—HLF-T cells stably infected with SV40 early region expressing retroviruses; HLF-TSR—HLF-TS cells stably infected with H-RasV12 expressing retroviruses; HCT116; and HCT116$^{DKO}$—HCT116 cells with genetic disruption of DNMT1 and DNMT3b (Rhee et al, 2002). Cells were maintained in DMEM (Gibco) containing 10% FBS (Gibco), 100 U/ml penicillin, and 100 μg/ml streptomycin at 37°C in a humidified incubator containing 5% $CO_2$. Conditions for SILAC labeling were reported previously (Grolimund et al, 2013; Majerska et al, 2017).

### Plasmids

Recombinant retroviruses were produced in HEK293T cells transfected with a pBABE expression vector (pBABE-hygro-hTERT [Cristofari & Lingner, 2006], pBABE-puro-HRasV12 [9051; Addgene], pBABE-neo-largeTgenomic [10891; Addgene]), and pcl-Ampho packaging vector (Naviaux et al, 1996). For validation of telomeric localization of candidate proteins by anti-HA ChIP, cDNAs from HLF-TSR cells were amplified by PCR and introduced by InFusion cloning (Clontech) into pcDNA6-derived vectors (Invitrogen) for expression as 3xHA-tagged proteins. The shRNA-expressing vectors were generated by cloning the corresponding double-stranded DNA oligonucleotides into pSuper vectors digested with BglII and HindIII. The targeting sequences are as follows: pSuper-blast-shSAMHD1 #2 5′-CGCATGCTGAAGCTAAGTA-3′; pSuper-blast-shSAMHD1 #4 5′-GTATCGCATTTCTACAGCA-3′; and pSuper-puro-shTRF1 5′-GAATATTTGGTGATCCAAA-3′.

### QTIP

QTIP experiments were carried out as described previously (Grolimund et al, 2013; Majerska et al, 2017) with the following modifications. 240–490 × 10$^6$ cells of each cell type were used per IP. Upon mixing the two cell types in a 1:1 ratio, cells in suspension were cross-linked for 10 min at 25°C using a combination of 1% formaldehyde and 2 mM ethylene glycol bis(succinimidyl succinate). Chromatin-enriched pellets were resuspended at 25 × 10$^6$ cells/ml

in LB3 buffer (10 mM Tris–HCl, pH 8.0; 200 mM NaCl; 1 mM EDTA–NaOH, pH 8.0; 0.5 mM EGTA–NaOH, pH 8.0; 0.1% w/v sodium deoxycholate; 0.25% w/v sodium lauryl sarcosinate; and EDTA-free protease inhibitor complex [Roche]) and sonicated for 12 min at 4°C using the Focused-ultrasonicator (Covaris E220; 12 × 12-mm glass tubes with AFA fiber; duty: 5.0, peak incident power: 140, cycles: 200, amplitude: 0, velocity: 0, and dwell: 0). Immunoprecipitation was performed using beads covalently coupled with non-specific IgGs (sc-2027; Santa Cruz Biotechnology) or beads coupled to affinity-purified anti-TRF1 (rabbit #714) and anti-TRF2 antibodies (rabbit #40) according to the previously published protocol (Aeby et al, 2016).

### Mass spectrometry analysis and data processing

LC-MS/MS analysis was performed as described previously (Aeby et al, 2016). Raw MS data were analyzed using MaxQuant software (Cox & Mann, 2008) with FDR < 0.01 at the level of both proteins and peptides. Peak lists were searched against the human Uniprot database, using Arg10 and Lys8 as heavy labels (multiplicity of 2). The mass tolerance was set to 7 ppm for precursor ions and MS/MS accuracy was set at 0.5 D. Enzyme was set to trypsin with up to two missed cleavages. Proteins and peptides (minimum six amino acids) were identified using a target-decoy approach with a reversed database. At least two (unique + razor) peptides were required for protein identification. A cutoff was set to 0.1 for posterior error probability.

TRF/IgG enrichment was determined using MS/MS spectral counts, which were calculated from the evidences of protein groups. For each protein group, we summed up the MS/MS counts for each experiment. At least three MS/MS spectral counts were required per condition. The missing values were imputed with pseudo-counts of 0.5. The normalization of the spectral counts was performed following the normalization schema in Scaffold (http://www.proteomesoftware.com/products/scaffold/). Further statistical analyses and graphical displays were performed in Perseus software version 1.5.3.2 (http://www.coxdocs.org/), using mean values of the forward and reverse QTIP replicates. The expression values and ratios were log$_2$-transformed and normalized by subtracting the median. Proteins enriched in TRF IP over control IgG IP (i.e., showing significant TRF/IgG ratios) were calculated separately for each QTIP as statistically significant outliers using a Significance A left-sided test ($P < 0.05$). Only proteins with significant TRF/IgG ratio in at least two of four QTIPs were considered "telomeric." Upon manual inspection, proteins HRG and SERPINC1 were removed from the list of telomeric proteins because they were light labeled in both label-swap experiments, indicating they were external contaminants. Significant SILAC ratios were determined for each QTIP (both, for inputs and TRF IPs) using a Significance B two-sided test (both sides, $P < 0.05$). Proteins with a significant SILAC ratio in at least one of the four QTIPs were considered "transformation responsive." To create heatmaps, hierarchical clustering was performed in Perseus based on average Euclidean distance of the log$_2$ mean SILAC ratios. The mass spectrometry proteomics data have been deposited to the ProteomeXchange Consortium via the PRIDE (Vizcaino et al, 2016) partner repository with the dataset identifier PXD010088.

## siRNA and plasmid transfection

The cells were transfected with siRNA pools (5 pmol of each siRNA per well of a six-well plate) by calcium phosphate precipitation twice with a 36-h interval and harvested 80–100 h after the first transfection. siRNA oligonucleotides were purchased from Qiagen: SAMHD1 (SI00710500, SI04137217, SI04189332, and SI04243673), DCLRE1B (SI00134757, SI02778692, and SI02778699), TMPO (SI03215730, SI04951156, and SI04951163), NPAT (SI00660814, SI04343157, and SI04355708), PARP9 (SI04136244, SI04196108, SI04212642, and SI04247285), PNUTS (SI00041811, SI00041832, SI03032666, and SI03088141), and SMCHD1 (SI00454664 and SI00454678). siGFP 5′-GCAGCACGACTTCTT-CAAGTT-3′ was synthesized at Microsynth. For transient over-expression of 3xHA-tagged proteins, HEK293T cells in a 15-cm tissue culture dish were transfected with 30 $\mu$g of plasmid DNA using calcium phosphate precipitation. Cells were harvested for ChIP analysis 42–48 h post-transfection. HeLa cells were transfected with pSuper-Puro and pSuper-Blast constructs using Lipofectamine 2000 (Invitrogen) according to the manufacturer's protocol. 1 $\mu$g/ml puromycin and 5 $\mu$g/ml blasticidin were added to the medium 1 d after the transfection, and selection was maintained for 5 d.

## Chromatin immunoprecipitation

HEK293T cells were cross-linked with 1% formaldehyde for 25 min, resuspended in LB3 buffer (20-30 × 10$^6$ cells/ml), and sonicated for 30 min at 4°C using a Focused-ultrasonicator as described in the QTIP protocol. The sonicated extracts were centrifuged at 4°C for 15 min at 20,000 $g$ and the supernatant was mixed with four volumes of ChIP dilution buffer and precleared for 1 h at 4°C with Sepharose 6B (Sigma-Aldrich) pre-blocked with yeast tRNA (0.5 mg/ml of beads). The precleared lysate corresponding to 2 × 10$^6$ cells was supplemented with 40 µl of yeast tRNA-blocked Protein G 50% bead slurry (GE Healthcare Life Sciences) and 3 $\mu$g of anti-HA antibody (ab9110; Abcam). After an overnight incubation at 4°C, the beads were washed as for QTIP. The DNA from input and IP samples was isolated and analyzed by dot blot hybridization as described previously (Grolimund et al, 2013).

## Northern blotting analysis of TERRA

RNA extraction was performed using RNeasy Mini kit (Qiagen) according to manufacturer's protocol. Northern blot TERRA analysis was performed as described previously (Azzalin et al, 2007). In brief, 10 $\mu$g of total RNA was separated by electrophoresis in a 1.2% formaldehyde agarose gel and blotted onto a nylon membrane (Hybond N+; GE Healthcare Life Sciences). Upon UV-cross-linking, the membrane was hybridized overnight with a [$\alpha^{32}$P]-dCTP–labeled telomeric probe at 50°C. After hybridization, the membrane was washed three times with a 1× SSC/0.5% SDS solution for 20 min at 60°C. Radioactive signals were detected using a phosphorimager (FLA-3000; Fujifilm).

## Standard curves for TERRA copy number determination

To determine the copy number of TERRA in the qPCR reactions, we constructed a standard curve for each TERRA based on a recombinant plasmid containing the desired subtelomere sequence. The subtelomeres were amplified from HeLa genomic DNA using Phusion Green High-Fidelity DNA polymerase and the primers listed in Table S4. The PCR products were isolated from the gel, purified using the Qiagen gel extraction kit, and cloned in the pCR4 BLUNT-TOPO vector using the Zero Blunt TOPO PCR cloning kit for sequencing. All the plasmids were sequenced to confirm the subtelomere sequence. To construct the standard curves for each TERRA, we made a 10-fold serial dilution of each plasmid ranging from 3 × 10$^8$ to 3 × 10$^{-1}$. The concentration of each plasmid was determined using Qubit 4 Fluorometer (Invitrogen). We calculated the copy number concentration using the following formula:

$$\text{DNA (copy)} = \frac{6.02 \times 10^{23} (\text{copies mol}^{-1}) \times \text{DNA amount (g)}}{\text{DNA length (bp)} \times 660\,(\text{g mol}^{-1}\,\text{bp}^{-1})}.$$

Each standard curve was performed in duplicate. The $C_T$ values were plotted against the logarithm of the plasmid copy number and the standard curve was generated by a linear regression of the points. The PCR amplification efficiency was calculated as a percentage from the slope of the curve using the following formula:

$$E = \left(10^{\frac{-1}{\text{slope}}} - 1\right) \times 100.$$

## RT–qPCR analysis of TERRA

RNA was isolated using RNeasy Mini kit (Qiagen) according to the manufacturer's protocol. The RNA was reversed-transcribed using the Invitrogen SuperScript III Reverse Transcriptase as previously described (Feretzaki & Lingner, 2017). The cDNA was diluted 1/2 for routine relative quantification qPCR or 1/2 and 1/5 for absolute quantification. qPCR runs included both a no-template control and a no-reverse transcription control. All the experiments were performed in three biological replicates. qPCR was performed as previously described (Feretzaki & Lingner, 2017). To determine and analyze the relative changes in TERRA expression between the samples, we applied the 2$^{-\Delta\Delta Ct}$ method. The number of TERRA copies in the qPCR or the cDNA template was calculated based on the standard curve.

## RT–qPCR analysis of depletion efficiency

3 $\mu$g of RNA was reverse-transcribed using SuperScript III RT (Invitrogen) with 150 ng random primers (Promega) and 500 ng oligo (dT)$_{15}$ primers (Promega). The qPCR was performed as previously described (Feretzaki & Lingner, 2017) using primers listed in Table S5.

## Telomeric FISH on metaphase chromosomes

Cells were treated with 0.05 $\mu$g/ml demecolcine for 2 h, harvested, and incubated in 0.056 M KCl at 37°C for 7 min. Swollen cells were

fixed in cold methanol/acetic acid (3:1) overnight and spread on glass slides. After ageing overnight, the slides were rehydrated in 1× PBS, fixed in 4% formaldehyde, and dehydrated in an ethanol series. The slides were allowed to air-dry before applying the hybridization solution (10 mM Tris–HCl, pH 7.4; 70% formamide; and 0.5% blocking reagent [Roche]) containing Cy3-OO-(CCCTAA)$_3$ PNA probe. The spreads were denatured at 80°C for 3 min and hybridized at RT for 3 h. The slides were washed twice for 15 min with 10 mM Tris–HCl (pH 7.4)/70% formamide, and three times for 5 min with 0.1 M Tris–HCl (pH 7.4)/0.15 M NaCl/0.08% Tween-20 with DAPI in the second wash. The slides were dehydrated in an ethanol series and mounted in Vectashield embedding medium. Images were acquired using a Zeiss Axioplan II microscope.

## Immunofluorescence FISH

Cells were grown on coverslips, fixed in 4% formaldehyde, and permeabilized for 5 min in 0.1% Triton X-100/0.02% SDS/1× PBS, and then pre-blocked for 10 min in 2% BSA/1× PBS and blocked for 45 min in 10% goat serum/2% BSA/1× PBS. Cells were then incubated with 0.5 µg/ml α-53BP1 antibody (NB100-304; Novus Biologicals) for 1 h, washed three times for 4 min in 2% BSA/1× PBS, and incubated with Alexa Fluor 633–conjugated secondary antibody (1:500, A-21070; Invitrogen) for 30 min. After three washes in PBS, stained cells were fixed in 4% formaldehyde for 5 min. Both antibodies were diluted in 10% goat serum/2% BSA/1× PBS, and all steps of the IF protocol were performed at RT. For telomeric FISH, the stained and fixed coverslips were dehydrated in an ethanol series, air-dried, and hybridized with the Cy3-OO-(CCCTAA)$_3$ PNA probe as described above. Images were captured using a Zeiss LSM 700 confocal microscope.

## Telomere Restriction Fragment (TRF) analysis

Genomic DNA was isolated using the Wizard Genomic DNA Purification kit (Promega) and subjected to restriction digestion with RsaI and HinfI, in the presence of RNase A, at 37°C overnight. Digested DNA was extracted using phenol–chloroform–isoamyl alcohol, followed by isopropanol precipitation. 1 µg of digested DNA was separated on 1% agarose gel by pulsed-field gel electrophoresis using a Bio-Rad CHEF DR-II apparatus or on 0.8% agarose gel by constant field gel electrophoresis. The gels were dried and hybridized with [α$^{32}$P]-dCTP–labeled telomeric probe at 50°C as described (Grolimund et al, 2013). Radioactive signals corresponding to single-stranded telomeric DNA were detected using a phosphorimager (FLA-3000; Fujifilm). Thereafter, the gel was denatured in 0.5 M NaOH, 1.5 M NaCl for 30 min, neutralized for 15 min with 0.5 M Tris–HCl, pH 7.5, and 1.5 M NaCl; and hybridized again to detect total telomeric DNA.

## Immunoblot analysis

Immunoblots were incubated separately with the following primary antibodies: anti-ATM pS1981 (1:1,000, ab81292; Abcam), anti-Chk2 pT68 (1:1,000, 2661; Cell Signaling), anti-p53 (1:1,000, sc-126; Santa Cruz), anti-p21 (1:5,000, ab109520; Abcam), anti-Actin (1:2,000, sc-1616; Santa Cruz), anti-TRF2 (1:1,000, 05-521; Millipore), and anti-

γH2AX (1:1,000, 05-636; Millipore), followed by a corresponding horseradish peroxidase–conjugated secondary antibody.

## Flow cytometry analysis

Following fixation in 70% ethanol, the pellet from $5 \times 10^5$ cells was resuspended in 250 µl of PBS containing 0.2 µg/ml RNAse A and incubated for 15 min at 37°C. The DNA was stained by adding 250 µl of 80 µg/ml of propidium iodide, followed by 10-min incubation at 4°C in the dark. 250 µl of PBS was added to each sample before FACS analysis on Accuri C6 (BD Biosciences). The percentage of cells in each phase of the cell cycle was determined using the Watson Pragmatic computational model in FlowJo software (TreeStar).

## DNA/RNA hybrid immunoprecipitation

$60–80 \times 10^6$ cells were harvested for each condition and lysed for 5 min in cold RLN buffer (50 mM Tris-Cl, pH 8.0; 140 mM NaCl; 1.5 mM MgCl$_2$; 0.5% v/v Nonidet P-40; 175 µl buffer per $10 \times 10^6$ cells) supplemented with 1 mM DTT and 100 U/ml RNasin Plus (Promega). After centrifugation (300 $g$, 2 min, 4°C), the nuclei-enriched pellet was resuspended in RLT buffer (RNeasy Plus kit; [Qiagen]; 500 µl buffer per $20 \times 10^6$ cell equivalent) supplemented with β-mercaptoethanol at 10 µl/ml and homogenized by passing through an insulin needle. Nucleic acids were thereafter isolated by a phenol:chloroform:isoamylalcohol (25:24:1; Biosolve BV) separation, followed by isopropanol precipitation. The nucleic acids were dissolved in H$_2$O (400 µl per $20 \times 10^6$ cell equivalent) and sonicated using a Focused-ultrasonicator (Covaris E220) to obtain DNA fragments below 500 bp. 150 µl of sonicated nucleic acids were treated for 90 min at 37°C with either 10 µl of RNase H (1 U/µl; Roche) or H$_2$O. The reaction was stopped by addition of 2 µl of 0.5 M EDTA. The samples were mixed with 1,738 µl of buffer 1 (10 mM Hepes-KOH, pH 7.5; 275 mM NaCl; 0.1% SDS; 1% Triton X-100; and 0.1% Na-deoxycholate) and precleared for 1 h at 4°C with 80 µl of 50% Protein G bead slurry (GE Healthcare Life Sciences) that had been pre-blocked for 1 h at 4°C with yeast tRNA (1 mg tRNA per 1 ml of 50% bead slurry). The IP reactions were set up containing 800 µl of precleared extract, 40 µl of tRNA-blocked Protein G beads, and 1 µg of S9.6 antibody (022715; Kerafast) or mouse IgG (sc-2025; Santa Cruz Biotechnology). The reactions were incubated for 90 min at 4°C on a rotating wheel and then washed consecutively for 5 min each time with 1 ml of buffer 2 (50 mM Hepes-KOH, pH 7.5; 140 mM NaCl; 1% Triton X-100; 0.1% Na-deoxycholate; and 1 mM EDTA), buffer 3 (50 mM Hepes-KOH, pH 7.5; 500 mM NaCl; 1% Triton X-100; 0.1% Na-deoxycholate; and 1 mM EDTA), buffer 4 (10 mM Tris–HCl, pH 8.0; 250 mM LiCl; 1% NP-40; 1% Na-deoxycholate; and 1 mM EDTA), and 1× TE buffer (10 mM Tris–HCl, pH 8.0, and 1 mM EDTA). The input and precipitated DNA was analyzed by dot blot hybridization.

## T-circle assay

T-circle assay was modified from the work of Zellinger et al (2007). 2 µg of genomic DNA was digested with 10 U/µg (each) HinfI and MboI (New England Biolabs) in CutSmart buffer in the presence of 10 µg/ml RNase A overnight at 37°C. 1.5 µg of cut DNA was ethanol-precipitated and resuspended in an annealing buffer (0.2 M Tris, pH

7.5; 0.2 M KCl; and 1 mM EDTA) with 1 $\mu$M (CCCTAA)$_3$ primer containing thiophosphate linkages between the three 3′ terminal nucleotides. The mix was denatured at 96°C for 5 min and cooled down to 25°C for 2 h. DNA was ethanol-precipitated and resuspended in 10 $\mu$l of H$_2$O for the T-circle reaction (10 $\mu$l sample combined with 10 $\mu$l 0.2 mg/ml BSA, 0.1% Tween, 1 mM each dATP, dGTP, dTTP, dCTP, 1× $\phi$29 buffer, and 7.5 U $\phi$29 DNA polymerase [New England Biolabs]). Primer extension was carried out at 30°C for 12 h. The $\phi$29 DNA polymerase was inactivated by incubation at 65°C for 20 min. The extension products were separated by agarose gel electrophoresis (0.6% agarose at 2 V/cm for 16 h). The gel was then dried at 50°C for 2 h, denatured and hybridized with a specific telomeric probe. U2OS genomic DNA of the ALT cell line was used as a positive control.

## Supplementary Information

## Acknowledgements

We thank Patricia Renck Nunes for providing the plasmids for TERRA copy number determination. We gratefully acknowledge Patrick Reichenbach for immortalization of HLFs. The mass spectrometry analysis was performed by the Proteomics Core Facility and Technology Platform at Ecole Polytechnique Fédérale de Lausanne (EPFL) (Florence Armand, Romain Hamelin, and Marc Moniatte). Members of the Lingner laboratory are thanked for discussions, technical advice, and sharing reagents. Research in J Lingner's laboratory was supported by the Swiss National Science Foundation (SNSF), the SNSF-funded NCCR RNA and disease network, an Initial Training Network grant (CodeAge) from the European Commission's Seventh Framework Programme (grant agreement no. 316354), the Swiss Cancer League, and EPFL. M Feretzaki was supported by a Marie Curie postdoctoral fellowship.

### Author Contributions

J Lingner: conceptualization, funding acquisition, and writing—original draft, review, and editing.

J Majerska: data curation, investigation, methodology, project administration, and writing—review and editing.

M Feretzaki: investigation.

G Glousker: investigation.

### Conflict of Interest Statement

The authors declare that they have no conflict of interest.

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
