## [Reviewer comments · Life Science Alliance]

Transformation-induced stress at telomeres is counteracted through changes in the telomeric proteome including SAMHD1

Jana Majerska, Marianna Feretzaki, Galina Glousker and Joachim Lingner
DOI: 10.26508/lisa.201800121

Review timeline:

Submission Date:	2 July 2018
Revision Received:	2 July 2018
Editorial Decision:	4 July 2018
Accepted:	5 July 2018

Report:

(Note: Letters and reports are not edited. The original formatting of letters and referee reports may not be reflected in this compilation.)

Please note that the manuscript was previously reviewed at another journal and the reports were taken into account in inviting a revision for publication at *Life Science Alliance* prior to submission to *Life Science Alliance*.

1st Editorial Decision

4 July 2018

Thank you for submitting your revised manuscript entitled "Transformation-induced stress at telomeres is counteracted through changes in the telomeric proteome".

The manuscript was previously reviewed at a different journal and the referee reports have been transferred to Life Science Alliance. You provided a revised manuscript and a detailed point-by-point response to the reports obtained during peer-review elsewhere. During our pre-transfer discussion, we had outlined that we would expect you to address the previously raised concerns by text changes, and that we would appreciate if you could revise your work to address the following:

- Ref#1: point 4: provide deep proteome coverage; discuss limitations or add whole lysate results if already at hand. Address point 5 and 7 (the latter by changing/clearly mention limited support for model) as well as minor points
- Ref#2: address the four points raised, point four can be simply commented on

We appreciate the way you addressed all these concerns and the detailed response to the three referees. We are thus happy to accept your manuscript in principle for publication in Life Science Alliance.

REFeree REPORTS OBTAINED DURING PEER REVIEW ELSEWHERE**Referee #1:**

In their manuscript "Transformation-induced stress at telomeres is counteracted through changes in the telomeric proteome including SAMHD1" Majerska et al. compare the composition of telomeric chromatin of normal and transformed human fibroblasts by ChIP-MS using SILAC labelling for quantitative proteomics and supplement the analysis of the telomeric proteins by an examination of TERRA levels and the abundance of R-loops at telomeres. In addition, they test by siRNA loss-of-function whether 7 of the proteins that were upregulated during transformation have an impact on

telomere fragility and show that co-depletion of TRF1 with one of these factors, SAMHD1, leads to "outsider" telomeres, i.e. telomeric signals that are completely detached from metaphase chromosomes.

The systematic and quantitative analysis of the telomeric chromatin composition in a cellular transformation model could be a rich resource for the telomere and cancer communities. However, the exact experimental design that was chosen here is somewhat unfortunate as it restricts the data to a subset of binary comparisons. Furthermore, as a proteomics resource, an expression analysis of the total protein levels would be an essential layer of information to be able to better interpret the data on telomere-chromatin associated proteins in the first place. In addition, the functional tests remain somewhat disconnected from the actual research questions, e.g. the function of SAMHD1 in relationship to TRF1 is not performed in the context of what the authors set out to study here. Therefore, both the validation of the identified proteins and their changes in telomeric abundance as well as the functional impact of these data in the context of oncogenic transformation are difficult to assess. Overall, this is a promising project that should be executed more systematically with more extensive and rigorous functional follow-up.

Major concerns:

1. QTIP is basically ChIP-MS using TRF1 and TRF2 antibodies. The authors should explain this in the introduction and cite relevant publications for ChIP-MS/RIME beyond their own work (e.g. PMID: 16284124; PMID: 23295261; PMID: 23403292). Naming the basic same approach differently only for TRF1/2 might create confusion as it implies some special technique that is somewhat different from performing ChIP-MS on any other DNA-binding protein, transcription factor etc. However, as a ChIP-MS approach it comes with the same technical limitations, e.g. it cannot distinguish whether a protein is co-enriched with TRF1/2 while binding to telomeric chromatin or to any of the unique genomic binding sites that have been reported e.g. for TRF2 related to its transcription factor activity.

2. While it is a strength of the manuscript that the authors used a quantitative approach for their LC-MS/MS analysis, the experimental design is unfortunate. SILAC labelling is indeed particularly useful for pull-down applications to control biases in the handling of different samples/conditions, but it is more or less restricted to binary combinations as carried out by the authors. The aim of this manuscript is to compare different stages of transformation. Therefore, it would be beneficial to allow a comparison of all four conditions against each other (and in particular to be able to compare all conditions against normal HLF, which only exists in comparison to HLF-TSR). Other quantitative MS solutions such as TMT labelling or label-free approaches (e.g. IBAQ with a sufficient amount of replicates) would be able to incorporate such a 4-way comparison and to include quantitative assessments of the background proteins (see also next point). This would make the analysis more comprehensive and valuable as a resource.

3. The use of SILAC for 2-way comparisons also impacts the identification of telomere-specific proteins vs. background proteins. Here, the authors rely on spectral counts from separate ChIP-MS reactions using IgG. Ideally, the authors would determine the proteins that are quantitatively enriched at telomeres in each condition separately prior to comparison between different experimental conditions. This could have been achieved by performing ChIP reactions with TRF1/2 and IgG on heavy and light extracts, respectively, in separate tubes and mixing these reactions at the very end/at the elution point. Again, TMT labelling or label-free approaches would allow for all of these comparisons to be quantitatively integrated into the entire dataset.

In this context, the determination of significant TRF/IgG ratios as well as significant SILAC ratios in TRF IPs is based on statistics using two replicates only ("mean values of the forward and reverse QTIP replicates" (page 16)), which raises concerns about the statistical robustness of how these protein lists were determined. To provide a very thorough reference resource, triplicate measurements for both 'forward' and 'reverse' reactions would have been ideal, preferably from 3 separate batches of HLF cells transduced with the various vectors (hTERT, SV40 and the H-RasV12 allele).

4. As the authors show in Fig. EV2B a possible explanation for any of the observed changes in the telomeric chromatin composition could be simply a change in protein expression levels as seen for TRF2. On Page 8 they state that they "analyzed the proteome of chromatin extracts of the four

different HLF lines". First of all, the corresponding figure legend in Fig. EV2B only mentions the comparison of HLF-TSR vs HLF-T, but even for this data only a GO term analysis of the putatively up- or down-regulated proteins seems to be provided. The authors should (i) provide deep proteome coverage quantitatively comparing all four experimental conditions against each other and (ii) perform these experiments on whole cell lysates as chromatin extracts bias the analysis. Ideally, total expression levels will allow to distinguish whether proteins that are bound more or less frequently to telomeric chromatin by ChIP-MS are simply up- or down-regulated for their expression level or whether binding to telomeres is regulated. This analysis is critical to assess whether the ChIP-MS data of this manuscript provides an additional layer of information beyond what RNA-seq and protein expression datasets would reveal. Chromatin extract proteomes or more localised binding tests to other regions of the genome would be beneficial to distinguish whether the observed effects are exclusive to telomeres or are rather part of a general adaptation that happens to include telomeres.

5. The functional data presented in Fig. 5A is uninterpretable. The authors assess telomere fragility phenotypes that are scored in two replicates upon transfection of a single siRNA. The authors merge various abnormalities such as signal-free-ends, telomeric doublets etc. into a single percentage, while these abnormalities would ideally be assessed individually. What is more, while fluctuations of +/-20-30% of these combined events are described it is not possible to deduce whether any of these changes are statistically significant. Furthermore, these values should be normalised to the number of metaphases analysed not to the total number of telomeres to avoid that events in individual cells disproportionately affect averages. Another concern in this context is that the authors state to have scored 1,100-4,800 telomeres per condition. With an expected 96 telomeres per cell this spans 10-50 metaphases. Even neglecting this broad range of individual cells analysed, data at the lower end of this span would seem too few. Parts of this criticism also applies to Fig. 6C/D/G/H.

In addition, the siPPP1R10 knock-down casts particular doubt on this data. Although this is the only case for which no substantial siRNA knock-down is achieved even on mRNA level, it scores for the highest increase of fragile telomeres in HLF-TSR cells and the second highest increase in HeLa cells. Overall, these data are incomplete even concerning validation on a technical level (knock-down levels by Western Blot; multiple siRNAs etc.).

6. While the data on SAMHD1 and the "outsider" phenotype (Fig. 6) is interesting this data is somewhat premature and feels rather patched on. The authors set out to determine changes in telomeric chromatin upon transformation but a potential function of SAMHD1 in telomere stability in HeLa cells seems quite detached from the purpose of this manuscript. How do SAMHD1 levels at telomeres relate to telomere fragility in transformed cells? Are they a cause or consequence? Do changes in SAMHD1 levels at telomeres promote/impair the transformation process? The authors reason that upregulation of proteins such as SAMHD1 protect against transformation-induced replicative stress/general fragility. However, this is only one possible explanation of the data and the authors should thus either provide functional evidence linking their proteomics identification more directly to their transformation protocol or consider validating the "outsider" phenotype in detail and to develop this into a separate study.

7. As the authors provide little mechanistic insights, the model in Fig. 7 seems premature and should be removed.

8. As a proteomics resource all raw data (ChIP-MS & expression proteomics) should be uploaded to an appropriate repository (e.g. PRIDE).

Minor concerns:

1. The authors state that Fig. EV1 shows that "telomeres in HLF with inactive telomerase shortened upon proliferation" (page 6). However, the figure legend does not state anything about passages of the individual samples and the blot only shows a single sample for HLF grown in heavy and light SILAC medium, respectively. The authors should add the data of telomere shortening in HLF vs. no shortening in the other conditions and add information on passages in the figure legend.

2. For the quantification of telomeric DNA enrichment in Fig. 1C/D the authors should provide images of the dot-blot (e.g. as EV figures). Furthermore, the data representation should be changed

to displaying the data points for the two replicates each separately instead of using bar plots with error bars. The use of error bars implies a data depth that is not backed by two replicates.

3. Related to major concern no. 3 the authors list spectral count enrichments for shelterin proteins in Fig. 1E. The authors report a fold change based on spectral counts. While spectral counts can be a semi-quantitative approximation the use of fold enrichments is misleading as those are not truly quantitative values. Either the authors simply report the spectral counts for TRF vs IgG as absolute values or (preferably) they should analyse this comparison with quantitative readouts.

4. The authors state that "the identified proteins overlapped partially with proteins that were identified in previous telomeric proteomic screens" (page 7). The authors should provide the data to this statement which is currently missing. In addition, the cited studies are a selection and the authors should include other relevant screens (e.g. PMID: 27829156; PMID: 28176777; PMID: 27676121; PMID: 28841410).

5. Fig. 2C/D states error bars corresponding to 2-3 technical replicates. The authors should generate 3 biological replicates for each condition as the bare minimum to substantiate the statistical evaluation. As mentioned above bar plots with error bars are in any case a suboptimal representation for two replicates and less. For the comparison between EV and CLP1 a statistical analysis seems to have been performed. It is not clear why this analysis would only exist for N-3xHA-CLP1 vs. EV but for none of the remaining conditions.

6. In a similar vein, the authors describe SAMHD1, TCEA1, ARHGAP1, FHL1 and PPP1R10 as novel telomere-associated proteins identified from HLF cells. However, the validation of this (Fig. 2C/D) suggests a very weak binding/enrichment at best in almost all conditions, for which the authors should provide images of the dot-blot. In addition, this validation has been carried out in HEK293T cells. Given the context of the manuscript a SV40-containing cell line is an unfortunate choice for validation and these experiments should ideally be repeated in HLF cells.

7. TERRA quantification in Fig. 4A should be performed in 3 biological replicates. Related to this in Fig. 4B, as above, it is unclear why only selected comparisons are indicated with stars or n.s. even for the same chromosome arm (e.g. 9p HLF-T is not annotated for the p-value).

8. In Fig. 5B the number of biological replicates for assessing the mRNA knock-down levels are not indicated.

9. Likewise in Fig. EV4 the number of replicates for quantification of TIFs nor the number of cells assayed are indicated. Given the lack of statistical analysis, it would be a concern if no replicates had been performed and/or only a small number of cells had been quantified. Ideally, the authors would report the frequency distribution of the number of TIFs per cell (e.g. 1, 2, 3,...) instead of using a sharp cut-off of >4/cell, which can create a bias.

10. Similar plots to Fig. 3A for all comparisons should be provided as a supplementary figure for better assessment of the data. Furthermore, plots comparing the log₂ 'forward' and 'reverse' SILAC ratios against each other for all experiments (both TRF IPs as well as IgGs) should be provided to assess the reproducibility of the data and to exclude labelling artefacts.

11. The data representation of Table EV1 "List of all identified proteins" would benefit from restructuring and is lacking key information that MaxQuant is providing:

a. Remove the column "Fasta headers". It takes a lot of space, but adds limited information about the reference fasta file that has been used. The essential information has been extracted by MaxQuant and is already found in the "Gene names" columns etc.

b. The authors use spectral counts to determine TRF/IgG enrichment, yet the spectral count information is not reported.

c. Since the authors used 'forward' and 'reverse' samples, the individual SILAC ratios should be reported. Currently there seems to be only the result from one direction and/or an averaged value presented (not clear which one it is).

d. The columns "Q-value", "Intensity", "Intensity L" and "Intensity H" (the latter two for 'forward' and 'reverse' experiments) should be represented.

12. Western Blots (Fig. 6, Fig. EV2) should include information on molecular weights and separately cropped lanes should remain clearly separate (e.g. Fig. 6A SAMHD1 and hnRNPA1 and TRF1 and hnRNPA1)

Non-essential suggestions:

1. SAMHD1 is mentioned very briefly in the introduction almost just like an add-on without any reasoning why this protein was chosen etc. In the discussion the authors mention SAMDH1 as one of the top hits and that "therefore we decided to explore its function at telomeres" (page 13). The flow of the manuscript would benefit from elaborating further on this both in the introduction and when transiting to the SAMDH1 data in the Results section.

2. The authors describe that "cell populations to be compared were mixed and cross linked" (page 6). From this description and the Fig 1A it appears that the authors first harvested cells and then cross linked the mixed SILAC pairs. This is somewhat unusual as ideally the cross-linking is carried out while the cells are still attached/growing on their cell culture plate to capture the chromatin composition during normal cell proliferation. While trypsinisation and the short-time delay before cross-linking might not have an impact, this remains a possibility. If there was a major technical hurdle that requires this particular order of sample preparation, a comment in the Material and Methods section would be helpful.

Referee #2:

In this manuscript, Majerska and colleagues present an unbiased proteomic analysis of the proteins present at telomeres during cellular transformation and oncogenic stress. Using human fibroblasts, they made pairwise comparisons of the telomeric proteome during different stages of transformation. This analysis revealed several novel telomere-associated proteins and identified a role for SAMHD1 in preventing telomere replication defects.

Overall, the authors present what will be a valuable resource to the community and the report that SAMHD1 is important for telomere replication will stimulate work in this area. The experiments are well carried out, and their interpretation is adequate. I have a few specific points that could be addressed, but this paper will make a useful contribution to the field.

Specific points

1- The idea that during transformation cells require additional factors to properly replicate telomeres is fascinating. The authors could test this hypothesis testing whether depletion of SAMHD1 has a more severe effect in transformed HLF-TSR cells compared to non-transformed HLF-T cells.

2- The outside telomere phenotype is interesting and suggests that depletion of TRF1 in the context of reduced levels of SAMHD1 should result in rapid telomere shortening. Is this the case?

3- It would be interesting to test whether cellular transformation would be impaired by inhibition of genes that support telomere replication such as SAMHD1.

4- It is unclear whether the "outside" phenotype was observed only upon SAMHD1 depletion or whether this is a common feature upon knockdown of the identified genes (Fig 5 C-E).

Referee #3:

In this manuscript, Lingner and colleagues interrogate the composition of telomeres during stagewise transformation of human cells. Specifically they used a transformation protocol involving telomerase, SV40 early region and HRAS and quantitatively measured the abundance and identity of telomere associated proteins. They found that the shelterin complex is increased, particularly after the introduction of the SV40 early region. Depletion of one novel candidate SAMHD1 led to telomere breakage events in cells deprived of TRF1.

These studies are carefully performed and understanding how telomeric chromatin is altered during cell transformation is an interesting area that has not been well studied. Although the experimental model used in these studies is somewhat artificial, it provides a useful way to parse different

alterations that lead to transformation.

The most striking changes were observed upon introduction of the SV40 early region. One question is whether the changes observed at telomeres are specific to telomeres or if they reflect changes in chromatin throughout the genome. It is clear that the focus of these studies is the telomere but SV40 large T or small t may have effects on chromatin and the changes observed may just be the telomeric version of these changes.

A second question is whether the observed changes are due to SV40 large or small t or if they are due to alterations in RB, p53 and PP2A. There are now experimental models of transformation in which these pathways are manipulated directly and it would be useful to determine if these changes are specific for SV40.

The finding that SAMHD1 is required for telomere maintenance in the setting of TRF1 loss is potentially interesting but the current work is incomplete. First what is the effect of depleting each of the candidates identified? Do the cells arrest or die? This information is necessary to understand the context of the telomere changes.

Second, does SAMHD1 bind telomeres or TRF1 directly? If SAMHD1 depletion prevents telomere repair, what is the nature of the "outsider."

The proposed model is reasonable; however, can the authors show that SAMHD1 interacts with the HR machinery or otherwise link it to the model?

1st Revision – authors' response

2 July 2018

Referee #1:

In their manuscript "Transformation-induced stress at telomeres is counteracted through changes in the telomeric proteome including SAMHD1" Majerska et al. compare the composition of telomeric chromatin of normal and transformed human fibroblasts by ChIP-MS using SILAC labelling for quantitative proteomics and supplement the analysis of the telomeric proteins by an examination of TERRA levels and the abundance of R-loops at telomeres. In addition, they test by siRNA loss-of-function whether 7 of the proteins that were upregulated during transformation have an impact on telomere fragility and show that co-depletion of TRF1 with one of these factors, SAMHD1, leads to "outsider" telomeres, i.e. telomeric signals that are completely detached from metaphase chromosomes.

The systematic and quantitative analysis of the telomeric chromatin composition in a cellular transformation model could be a rich resource for the telomere and cancer communities. However, the exact experimental design that was chosen here is somewhat unfortunate as it restricts the data to a subset of binary comparisons. Furthermore, as a proteomics resource, an expression analysis of the total protein levels would be an essential layer of information to be able to better interpret the data on telomere- chromatin associated proteins in the first place. In addition, the functional tests remain somewhat disconnected from the actual research questions, e.g. the function of SAMHD1 in relationship to TRF1 is not performed in the context of what the authors set out to study here. Therefore, both the validation of the identified proteins and their changes in telomeric abundance as well as the functional impact of these data in the context of oncogenic transformation are difficult to assess. Overall, this is a promising project that should be executed more systematically with more extensive and rigorous functional follow-up.

Major concerns:

1. QTIP is basically ChIP-MS using TRF1 and TRF2 antibodies. The authors should explain this in the introduction and cite relevant publications for ChIP-MS/RIME beyond their own work (e.g. PMID: 16284124; PMID: 23295261; PMID: 23403292). Naming the basic same approach differently only for TRF1/2 might create confusion as it implies some special technique that is somewhat different from performing ChIP-MS on any other DNA-binding protein, transcription factor etc. However, as a ChIP-MS approach it comes with the same technical limitations, e.g. it cannot distinguish whether a protein is co-enriched with TRF1/2 while binding to telomeric

chromatin or to any of the unique genomic binding sites that have been reported e.g. for TRF2 related to its transcription factor activity.

Reply: The QTIP technology and its significant advantages have been discussed extensively in (Grolimund et al. 2013 and Majerska et al. 2017). The reader is referred to both of these papers in the introduction. Significantly, QTIP allows identification of the telomeric proteome in vivo. Through the implementation of SILAC, QTIP could for the first time provide a quantitative comparison of telomeric protein composition in cells with different telomeric states. Therefore, in the current study, QTIP allowed to define the changes of the telomeric proteome during oncogenesis. The papers the referee suggest to cite are not relevant to the development of QTIP nor to telomeres.

TRF2 is known to bind additionally to few non-telomeric sites in the genome, such as a subset of interstitial telomeric sequences (ITS), satellite repeats (PMID: 21423270 and PMID:21423278), and to few promoters involved in angiogenesis (PMID: 25437559) and neural differentiation (PMID: 24740933). However, these sites are minor in comparison to telomeres. In Figure 1C we employed a Dot-Blot analysis to measure the telomeric enrichment over the rest of the genome demonstrating high specificity and efficient telomeric pulldown. In addition we know that the large majority of TRF2 is present at telomeres as the TRF-RAP1 interaction stabilizes TRF2 at telomeres (PMID: 23086976). Considering that the length of the telomere in our cell lines is around ~12 KB, which binds thousands of copies of TRF1/TRF2, we are confident that proteins in close proximity to the few TRF2 non- telomeric binding sites will not correspond to a significant fraction. In agreement with this, we confirmed telomere association for several of the candidates in targeted ChIP experiments.

2. While it is a strength of the manuscript that the authors used a quantitative approach for their LC-MS/MS analysis, the experimental design is unfortunate. SILAC labelling is indeed particularly useful for pull-down applications to control biases in the handling of different samples/conditions, but it is more or less restricted to binary combinations as carried out by the authors. The aim of this manuscript is to compare different stages of transformation. Therefore, it would be beneficial to allow a comparison of all four conditions against each other (and in particular to be able to compare all conditions against normal HLF, which only exists in comparison to HLF-TSR). Other quantitative MS solutions such as TMT labelling or label-free approaches (e.g. IBAQ with a sufficient amount of replicates) would be able to incorporate such a 4-way comparison and to include quantitative assessments of the background proteins (see also next point). This would make the analysis more comprehensive and valuable as a resource.

Reply: As mentioned by the referee, the major strength of SILAC (but not TMT) is that it allows cell mixing immediately after cell harvest. This avoids possible artifacts caused by unequal sample manipulation. On the contrary, isobaric labeling is typically performed on desalted peptide samples, i.e. the ChIP and initial sample preparation for MS has to be performed separately for each cell line. The QTIP workflow entails several complicated sample processing steps, which if done in parallel before sample mixing, can introduce considerable variation and lead to error in quantitation. Thus we consider SILAC as the method of choice. Apart from that, at the time this project was initiated (2013), TMT labeling and label-free quantitative approaches were not well established yet.

3. The use of SILAC for 2-way comparisons also impacts the identification of telomere-specific proteins vs. background proteins. Here, the authors rely on spectral counts from separate ChIP-MS reactions using IgG. Ideally, the authors would determine the proteins that are quantitatively enriched at telomeres in each condition separately prior to comparison between different experimental conditions. This could have been achieved by performing ChIP reactions with TRF1/2 and IgG on heavy and light extracts, respectively, in separate tubes and mixing these reactions at the very end/at the elution point. Again, TMT labelling or label-free approaches would allow for all of these comparisons to be quantitatively integrated into the entire dataset.

In this context, the determination of significant TRF/IgG ratios as well as significant SILAC ratios in TRF IPs is based on statistics using two replicates only ("mean values of the forward and reverse QTIP replicates" (page 16)), which raises concerns about the statistical robustness of how these protein lists were determined. To provide a very thorough reference resource, triplicate measurements for both 'forward' and 'reverse' reactions would have been ideal, preferably from 3 separate batches of HLF cells transduced with the various vectors (hTERT, SV40 and the H-RasV12 allele).

Reply: TMT labeling or label-free approaches were not applicable when the project was started (see also response to major concern #2). The presented QTIP data come from 16 large-scale immunoprecipitation reactions, for which we used more than 3600 15-cm dishes of adherent cells, as well as large quantities of antibodies and other costly reagents. The proposed "triplicate measurement for both forward and reverse reactions" would mean 48 reactions and >10'800 tissue culture dishes, which is beyond feasibility. Even though each QTIP was performed only in two biological replicates (forward and reverse reactions), we provide robust evidence of TRF/IgG enrichment. Indeed, we have for each telomeric protein candidate 8 TRF/IgG values (4 QTIPs x 2 replicates). In addition, we detected for the majority of the proteins reproducible telomere association across all four cell lines. We considered only proteins with significant TRF/IgG ratio in at least two out of four QTIPs as putatively "telomeric" in our study.

4. As the authors show in Fig. EV2B a possible explanation for any of the observed changes in the telomeric chromatin composition could be simply a change in protein expression levels as seen for TRF2. On Page 8 they state that they "analyzed the proteome of chromatin extracts of the four different HLF lines". First of all, the corresponding figure legend in Fig. EV2B only mentions the comparison of HLF-TSR vs HLF-T, but even for this data only a GO term analysis of the putatively up- or down-regulated proteins seems to be provided. The authors should (i) provide deep proteome coverage quantitatively comparing all four experimental conditions against each other and (ii) perform these experiments on whole cell lysates as chromatin extracts bias the analysis. Ideally, total expression levels will allow to distinguish whether proteins that are bound more or less frequently to telomeric chromatin by CHIP-MS are simply up- or down-regulated for their expression level or whether binding to telomeres is regulated. This analysis is critical to assess whether the CHIP-MS data of this manuscript provides an additional layer of information beyond what RNA-seq and protein expression datasets would reveal. Chromatin extract proteomes or more localised binding tests to other regions of the genome would be beneficial to distinguish whether the observed effects are exclusive to telomeres or are rather part of a general adaptation that happens to include telomeres.

Reply: First, we would like to briefly discuss the proposed idea that "a possible explanation for any of the observed changes in the telomeric chromatin composition could be simply a change in protein expression levels". There are several mechanisms that can lead to upregulation of a protein at telomeres. Increased protein expression is one mechanism, which per se we consider not as uninteresting as the referee appears to suggest. This would apply to the cases where protein expression is the only/major limiting factor for telomere association. In addition, there are other mechanisms that regulate protein recruitment to telomeres, including binding site availability, post-translational modifications, cell-cycle phase, nuclear import/export, presence of a cofactor or an inhibitor, etc. Moreover, with the exception of telomere-specific proteins such as TRF1 and TRF2, it is unlikely that the entire protein pool is recruited to telomeres. For example, it has been demonstrated that the chromatin-bound fraction of shelterin RAP1 constitutes only ~50% of the total RAP1 protein in a cell, with the remaining protein localizing to cytoplasm (doi: 10.1074/jbc.M109.038026).

For the hypothetical cases where protein expression determines the number of molecules associated with telomeres, one can speculate that the expression is upregulated in response to an increased demand of transformed telomeric chromatin. All in all, we expect that different mechanisms may apply to factors that are identified here.

Nevertheless, we performed a comparison of chromatin extract proteomes for HLF-derived cell lines (Table S1 - List A labeled as SILAC ratio in input), which enabled identification of a few cases where the protein behavior in the telomeric fraction (TRF IP) appears distinct from the behavior in the bulk chromatin fraction (Input): see for example SAMHD1, SMCHD1, MRE11A, MSH6.

However, as can be seen in Table S1 - Lists B-C, many of the proteins of interest are low-abundant, and could not be reliably quantified in bulk chromatin extracts (incl. TRF1/TERF1, TPP1/ACD, Apollo/DCLRE1B, PKP4, GOPC, PCF11). The most straightforward explanation is that higher-abundance proteins masked their detection. Thus, in more complex samples, such as whole-cell lysates, their detection and quantification would be even more problematic. Indeed, the recently published deep proteome analysis of HeLa cells (doi: 10.1016/j.cels.2017.05.009), which might be "the deepest proteome of a human single-cell type to date", reports very low sequence coverage for many of these proteins (see Table S3). This substantiates the need to isolate telomeric chromatin prior to mass spectrometry in order to reduce complexity of the sample, increase the depth of

analysis and allow reliable identification of quantitative differences in telomeric protein composition.

5. The functional data presented in Fig. 5A is uninterpretable. The authors assess telomere fragility phenotypes that are scored in two replicates upon transfection of a single siRNA. The authors merge various abnormalities such as signal-free-ends, telomeric doublets etc. into a single percentage, while these abnormalities would ideally be assessed individually. What is more, while fluctuations of +/-20-30% of these combined events are described it is not possible to deduce whether any of these changes are statistically significant. Furthermore, these values should be normalised to the number of metaphases analysed not to the total number of telomeres to avoid that events in individual cells disproportionately affect averages. Another concern in this context is that the authors state to have scored 1,100-4,800 telomeres per condition. With an expected 96 telomeres per cell this spans 10- 50 metaphases. Even neglecting this broad range of individual cells analysed, data at the lower end of this span would seem too few. Parts of this criticism also applies to Fig. 6C/D/G/H. In addition, the siPPP1R10 knock-down casts particular doubt on this data. Although this is the only case for which no substantial siRNA knock-down is achieved even on mRNA level, it scores for the highest increase of fragile telomeres in HLF-TSR cells and the second highest increase in HeLa cells. Overall, these data are incomplete even concerning validation on a technical level (knock-down levels by Western Blot; multiple siRNAs etc.).

Reply: We have corrected the legend in Fig. 5 to clarify that we used siRNA pools, not individual siRNAs. In Figures 5C and 6C, we now present only data for typical telomere fragility (smears and multiple telomeric signals), whereas Tables S2 and S3 provide detailed analyses of all abnormalities. In the charts, we plot % of fragile/outside telomeres per metaphase and indicate statistical significance. Please note, that the HeLa cell line is hyper-triploid with most cells having 76–80 chromosomes; doi: 10.1534/g3.113.005777. The number of scored telomeres and metaphases (reported in Tables S2 and S3) is in the range of previously published studies (DOI 10.1016/j.cell.2009.06.021, doi: 10.1101/gad.543509).

Fig 5B: To validate candidate depletion, we report mRNA levels for all plotted experiments. We obtained comparable results using two different primer pairs per target (only one of them was used for the figure), confirming RT-qPCR specificity and sensitivity. For each siRNA screen replicate, we also confirmed depletion of SAMHD1 and SMCHD1 by western blotting (see the image below). Despite repeated trials, we have not reached better depletion of PNUTS. As we state in the text (page 10), PNUTS depletion strongly impacted on cell viability, which is in agreement with previously published reports of increased apoptosis and reduced proliferation of cancer cells upon PNUTS knockdown (doi: 10.4161/cbt.7.6.5839, doi: 10.1158/0008-5472.CAN-12-1394). We suspect that this may have lead to survival advantage and outgrowth of cells with higher PNUTS expression.

Efficient depletion of SAMHD1 and SMCHD1 using siRNA pools. Immunoblotting analysis of knockdown efficiency in HLF-TSR and HeLa cells used in Figure 5. HnRNPA1 is shown as a loading control.

6. While the data on SAMHD1 and the "outsider" phenotype (Fig. 6) is interesting this data is somewhat premature and feels rather patched on. The authors set out to determine changes in telomeric chromatin upon transformation but a potential function of SAMHD1 in telomere stability in HeLa cells seems quite detached from the purpose of this manuscript. How do SAMHD1 levels at telomeres relate to telomere fragility in transformed cells? Are they a cause or consequence? Do changes in SAMHD1 levels at telomeres promote/impair the transformation process? The authors reason that upregulation of proteins such as SAMHD1 protect against transformation-induced replicative stress/general fragility. However, this is only one possible explanation of the data and the authors should thus either provide functional evidence linking their proteomics identification more

directly to their transformation protocol or consider validating the "outsider" phenotype in detail and to develop this into a separate study.

7. As the authors provide little mechanistic insights, the model in Fig. 7 seems premature and should be removed.

Reply: Discovery of the outsider phenotype helped us building a testable model, which should inspire future work in several laboratories. We agree that the model is hypothetical at this stage and this is clearly indicated in the text, which we slightly revised to further emphasize this point. We have not been able to provide additional direct evidence at this time. However, several findings in support of this model have already been reported in literature which are cited.

8. As a proteomics resource all raw data (ChIP-MS & expression proteomics) should be uploaded to an appropriate repository (e.g. PRIDE).

Reply: The mass spectrometry proteomics data have been deposited to the ProteomeXchange Consortium via the PRIDE partner repository with the dataset identifier PXD010088.

Minor concerns:

1. The authors state that Fig. EV1 shows that "telomeres in HLF with inactive telomerase shortened upon proliferation" (page 6). However, the figure legend does not state anything about passages of the individual samples and the blot only shows a single sample for HLF grown in heavy and light SILAC medium, respectively. The authors should add the data of telomere shortening in HLF vs. no shortening in the other conditions and add information on passages in the figure legend.

Reply: The figure shows telomere length at the time of harvest for QTIP. We have not studied telomere length dynamics in this study but previously. Therefore, we have corrected the sentence to read as follows: " Upon large-scale cell expansion, HLF-T, HLF- TS and HLF-TSR were indistinguishable with regard to their telomere length whereas the telomeres in HLF with inactive telomerase were markedly shorter (Fig S1), due to progressive telomere attrition (PMID: 16424902)."

2. For the quantification of telomeric DNA enrichment in Fig. 1C/D the authors should provide images of the dot-blot (e.g. as EV figures). Furthermore, the data representation should be changed to displaying the data points for the two replicates each separately instead of using bar plots with error bars. The use of error bars implies a data depth that is not backed by two replicates.

Reply: The original Fig 1C-D has been replaced with a revised Fig 1C, in which data for the forward and reverse replicates is plotted separately. The images of dot-blot are provided in Fig S3.

3. Related to major concern no. 3 the authors list spectral count enrichments for shelterin proteins in Fig. 1E. The authors report a fold change based on spectral counts. While spectral counts can be a semi-quantitative approximation the use of fold enrichments is misleading as those are not truly quantitative values. Either the authors simply report the spectral counts for TRF vs IgG as absolute values or (preferably) they should analyse this comparison with quantitative readouts.

Reply: We now provide the actual MS/MS spectral counts for the TRF IP and IgG IP (see Fig 1D). The data shows the very specific enrichment of shelterin proteins using TRF1 and TRF2 antibodies and complete absence of non-specific capture of telomeric proteins on IgG beads.

4. The authors state that "the identified proteins overlapped partially with proteins that were identified in previous telomeric proteomic screens" (page 7). The authors should provide the data to this statement which is currently missing. In addition, the cited studies are a selection and the authors should include other relevant screens (e.g. PMID: 27829156; PMID: 28176777; PMID: 27676121; PMID: 28841410).

Reply: We have added several examples of the proteins that were identified in our screen as well in previous telomeric proteomic screens on page 7 of the manuscript. One major advance of QTIP is the analysis of telomeric chromatin in live cells. Therefore, we are not citing the in vitro studies that

are suggested by the referee as they are prone to artifacts and not comparable to the studies on telomeric chromatin in live cells. However, we do cite the high quality studies that employed an alternative ChIP-MS based approach (PiCh) which is most comparable to QTIP.

5. Fig. 2C/D states error bars corresponding to 2-3 technical replicates. The authors should generate 3 biological replicates for each condition as the bare minimum to substantiate the statistical evaluation. As mentioned above bar plots with error bars are in any case a suboptimal representation for two replicates and less. For the comparison between EV and CLP1 a statistical analysis seems to have been performed. It is not clear why this analysis would only exist for N-3xHA-CLP1 vs. EV but for none of the remaining conditions.

Reply: We replaced Fig 2C-D with a revised Fig 2C. We have reanalyzed the data by subtracting the background intensity of the corresponding empty vector controls (negative controls for non-specific binding) in order to distinguish true signal from the background noise. We plotted side-by-side telomeric and Alu-repeat DNA recovery and the numbers above the columns indicate which proteins are enriched at telomeres compared to Alu repeats. We have not done statistical analysis as instead of biological replicates, we performed independent experiments with N- and C- terminally tagged proteins, which both localized to telomeres.

6. In a similar vein, the authors describe SAMHD1, TCEA1, ARHGAP1, FHL1 and PPP1R10 as novel telomere-associated proteins identified from HLF cells. However, the validation of this (Fig. 2C/D) suggests a very weak binding/enrichment at best in almost all conditions, for which the authors should provide images of the dot-blot. In addition, this validation has been carried out in HEK293T cells. Given the context of the manuscript a SV40-containing cell line is an unfortunate choice for validation and these experiments should ideally be repeated in HLF cells.

Reply: We have corrected the figure as specified in reply to point #5. The images of dot-blot are provided in Fig. S4. The HEK293T cell line is routinely used for studies that involve ectopic protein expression. We do not see why the presence of SV40 would discredit the observation that these proteins localize to telomeres. The detection of the candidates at telomeres by ChIP is clearly demonstrated. The not more substantial precipitation of telomeric DNA may be due to the indirect association of these proteins with the DNA.

7. TERRA quantification in Fig. 4A should be performed in 3 biological replicates. Related to this in Fig. 4B, as above, it is unclear why only selected comparisons are indicated with stars or n.s. even for the same chromosome arm (e.g. 9p HLF-T is not annotated for the p-value).

Reply: In Fig 4A, we included a second biological replicate. Furthermore, the transformation-induced increase in TERRA expression was observed using two technical approaches (Northern blotting in 2 biological replicates, RT-qPCR in 3 biological replicates) and is therefore well supported. In Fig 4B, we now show statistical significance for all samples. As specified in the figure legend, bars lacking * are not significant.

8. In Fig. 5B the number of biological replicates for assessing the mRNA knock-down levels are not indicated.

Reply: The figure legend has been corrected.

9. Likewise in Fig. EV4 the number of replicates for quantification of TIFs nor the number of cells assayed are indicated. Given the lack of statistical analysis, it would be a concern if no replicates had been performed and/or only a small number of cells had been quantified. Ideally, the authors would report the frequency distribution of the number of TIFs per cell (e.g. 1, 2, 3,...) instead of using a sharp cut-off of >4/cell, which can create a bias.

Reply: In the revised figure (Fig S8), we now plot the frequency distribution and indicate the number of analyzed cells.

10. Similar plots to Fig. 3A for all comparisons should be provided as a supplementary figure for better assessment of the data. Furthermore, plots comparing the log2 'forward' and 'reverse' SILAC

ratios against each other for all experiments (both TRF IPs as well as IgGs) should be provided to assess the reproducibility of the data and to exclude labelling artefacts.

Reply: These plots are now included as supplementary figures (Fig S3 and Fig S5). We have inspected the behavior of the 136 putative telomeric proteins for potential labeling artifacts. As stated in the Materials and Methods section, we excluded 2 proteins from the list as potential external contaminants.

11. The data representation of Table EV1 "List of all identified proteins" would benefit from restructuring and is lacking key information that MaxQuant is providing:

- Remove the column "Fasta headers". It takes a lot of space, but adds limited information about the reference fasta file that has been used. The essential information has been extracted by MaxQuant and is already found in the "Gene names" columns etc.
- The authors use spectral counts to determine TRF/IgG enrichment, yet the spectral count information is not reported.
- Since the authors used 'forward' and 'reverse' samples, the individual SILAC ratios should be reported. Currently there seems to be only the result from one direction and/or an averaged value presented (not clear which one it is).
- The columns "Q-value", "Intensity", "Intensity L" and "Intensity H" (the latter two for 'forward' and 'reverse' experiments) should be represented.

Reply: We have removed the Fasta headers. The requested proteomics data can now be accessed via the PRIDE repository.

12. Western Blots (Fig. 6, Fig. EV2) should include information on molecular weights and separately cropped lanes should remain clearly separate (e.g. Fig. 6A SAMHD1 and hnRNPA1 and TRF1 and hnRNPA1)

Reply: We have added molecular weight markers and frames, clearly separating the cropped membranes.

Non-essential suggestions:

1. SAMHD1 is mentioned very briefly in the introduction almost just like an add-on without any reasoning why this protein was chosen etc. In the discussion the authors mention SAMDH1 as one of the top hits and that "therefore we decided to explore its function at telomeres" (page 13). The flow of the manuscript would benefit from elaborating further on this both in the introduction and when transiting to the SAMDH1 data in the Results section.

2. The authors describe that "cell populations to be compared were mixed and cross linked" (page 6). From this description and the Fig 1A it appears that the authors first harvested cells and then cross linked the mixed SILAC pairs. This is somewhat unusual as ideally the cross-linking is carried out while the cells are still attached/growing on their cell culture plate to capture the chromatin composition during normal cell proliferation. While trypsinisation and the short-time delay before cross-linking might not have an impact, this remains a possibility. If there was a major technical hurdle that requires this particular order of sample preparation, a comment in the Material and Methods section would be helpful.

Reply: We favor crosslinking of cells in suspension for three main reasons. First, for a single QTIP replicate, we routinely harvest >400 tissue culture dishes. At this scale, on-plate fixation is impractical considering manipulation time, costs and potential health hazards. Second, harvesting the cells prior to crosslinking allows subsequent sample concentration and more precise cell counting which is required for proper cell mixing. Third, crosslinking the cells in solution after cell mixing ensures that the two cell populations are treated equally throughout the protocol.

We have modified the Material and Methods section to state the following: "Upon mixing the two cell types in a 1:1 ratio, cells in suspension were crosslinked for 10 min at 25 °C using a combination of 1% formaldehyde and 2 mM EGS." We also refer our readers to the previous QTIP papers (doi: 10.1038/ncomms3848; doi: 10.1016/j.ymeth.2016.08.003), which describe the protocol in a greater detail.

Referee #2:

In this manuscript, Majerska and colleagues present an unbiased proteomic analysis of the proteins present at telomeres during cellular transformation and oncogenic stress. Using human fibroblasts, they made pairwise comparisons of the telomeric proteome during different stages of transformation. This analysis revealed several novel telomere-associated proteins and identified a role for SAMHD1 in preventing telomere replication defects.

Overall, the authors present what will be a valuable resource to the community and the report that SAMHD1 is important for telomere replication will stimulate work in this area. The experiments are well carried out, and their interpretation is adequate. I have a few specific points that could be addressed, but this paper will make a useful contribution to the field.

Specific points

1- The idea that during transformation cells require additional factors to properly replicate telomeres is fascinating. The authors could test this hypothesis testing whether depletion of SAMHD1 has a more severe effect in transformed HLF-TSR cells compared to non-transformed HLF-T cells.

Reply: Although we repeatedly tried to address this point using two different protocols, we were not able to obtain metaphase chromosome spreads from HLF-T cells of sufficient quality to perform the experiment.

2- The outside telomere phenotype is interesting and suggests that depletion of TRF1 in the context of reduced levels of SAMHD1 should result in rapid telomere shortening. Is this the case?

Reply: Based on the analysis of stained telomeres in metaphase spreads, we can conclude that the breakpoint is close to the subtelomere-telomere boundary. Thus, the broken telomere is expected to have almost the same size as the normal TRF fragment. Consistently, we have not observed telomere shortening using TRF assay in short-term experiments (Fig S9A). Long-term experiments were at this stage not possible yet because of the essentiality of TRF1.

3- It would be interesting to test whether cellular transformation would be impaired by inhibition of genes that support telomere replication such as SAMHD1.

Reply: We agree that it would be very interesting to perform such experiments. However, we faced technical difficulties to partially inhibit gene expression precisely to the levels observed in non-transformed cells. Finally, in our opinion this analysis goes also somewhat beyond the scope of this paper.

4- It is unclear whether the "outside" phenotype was observed only upon SAMHD1 depletion or whether this is a common feature upon knockdown of the identified genes (Fig 5 C-E).

Reply: We do not expect the "outside" phenotype to be a common phenomenon as the identified candidates have quite diverse functions. Though not yet tested extensively with the here-identified factors, we already tested co-depletion of TRF1 with a number of other telomeric proteins obtained in another proteomic screen. From these experiments it seems that the "outsider" phenotype is very rare (likely specific to one DNA damage repair pathway).

Referee #3:

In this manuscript, Lingner and colleagues interrogate the composition of telomeres during stagewise transformation of human cells. Specifically they used a transformation protocol involving telomerase, SV40 early region and HRAS and quantitatively measured the abundance and identity of telomere associated proteins. They found that the shelterin complex is increased, particularly after the introduction of the SV40 early region. Depletion of one novel candidate SAMHD1 led to telomere breakage events in cells deprived of TRF1.

These studies are carefully performed and understanding how telomeric chromatin is altered during cell transformation is an interesting area that has not been well studied. Although the experimental

model used in these studies is somewhat artificial, it provides a useful way to parse different alterations that lead to transformation.

The most striking changes were observed upon introduction of the SV40 early region. One question is whether the changes observed at telomeres are specific to telomeres or if they reflect changes in chromatin throughout the genome. It is clear that the focus of these studies is the telomere but SV40 large T or small t may have effects on chromatin and the changes observed may just be the telomeric version of these changes.

Reply: In order to partly address this question, we quantified the changes in chromatin-enriched extracts, which were used as input for QTIP experiments. Unfortunately, due to technical limitations, we failed to reliably quantify several low-abundant proteins of interest (see also the response to Ref. #1 and quantitative data in Table S1). Recent improvement in sensitivity and reduction in costs of deep proteome analyses will likely facilitate more comprehensive analysis in the future (incl. parallel comparison of several genomic loci and the use of several models of transformation as suggested below). However, we respectfully note that such experiments would be beyond the time frame and scope of this study.

Even though it is expected that non-telomeric chromatin may also be affected during transformation, the alterations at telomeres appear crucial for genome integrity, especially considering their difficult-to-replicate and fragile nature.

A second question is whether the observed changes are due to SV40 large or small t or if they are due to alterations in RB, p53 and PP2A. There are now experimental models of transformation in which these pathways are manipulated directly and it would be useful to determine if these changes are specific for SV40.

Reply: We agree that this is an important and interesting question, which merits further investigation. However, we respectfully note that it is outside the scope of this study.

The finding that SAMHD1 is required for telomere maintenance in the setting of TRF1 loss is potentially interesting but the current work is incomplete. First what is the effect of depleting each of the candidates identified? Do the cells arrest or die? This information is necessary to understand the context of the telomere changes.

Reply: Our short-term knockdown studies show increased fragility upon depletion which is indicative of incomplete telomere replication. Fragile telomeres are known to induce ATR signaling which leads to cell cycle arrest in dependency of the genetic context. We consider the effects on telomere replication as the most significant result as responses to checkpoint activation have been characterized elsewhere.

Second, does SAMHD1 bind telomeres or TRF1 directly?

Reply: To validate SAMHD1 as a bona fide telomeric protein, we performed chromatin immunoprecipitation experiments against overexpressed 3xHA-tagged SAMHD1 (Fig 2C) as well as endogenous SAMHD1 (Fig below) in HEK293T and HLF-TSR cell lines, respectively. In both cases, we detected binding of SAMHD1 to telomeric chromatin, but no significant recruitment to Alu-repeat DNA.

To test the possibility that SAMHD1 forms a complex with shelterin proteins TRF1 and TRF2, exogenous 3xHA-tagged SAMHD1 was overexpressed together with 3xFlag-tagged TRF1 or Flag-TRF2 in the HEK293T cell line (Fig C below). Immunoprecipitation experiments with an anti-HA antibody confirmed the interaction of 3xHA-SAMHD1 with both, 3xFlag-tagged TRF1 and Flag-TRF2, even though the Flag-tagged proteins showed some non-specific binding to the affinity resin, likely as a consequence of gross overexpression of the constructs. The SAMHD1-TRF1 interaction was insensitive to benzonase treatment (Fig D below), making it unlikely that the interaction is DNA-dependent. These results open the possibility that SAMHD1 may be recruited to telomeres via direct interaction with shelterin, although SAMHD1 binding to telomeric DNA or recruitment via TERRA cannot be excluded, since it has been demonstrated that SAMHD1 is a nucleic-acid binding protein (DOI: 10.1074/jbc.M112.431148; DOI: 10.1093/nar/gkv633). Shelterin-independent mode of SAMHD1 recruitment to telomeres is supported by the QTIP4 experiment, in which SAMHD1

was upregulated at telomeres in HLF-TS cells compared to HLF-T, whereas TRF1 and TRF2 remained unchanged (Fig S5-QTIP4).

SAMHD1 localizes to telomeres and might form a complex with TRF1 and TRF2. (A) Validation of telomeric recruitment of SAMHD1 by ChIP against endogenous SAMHD1 in HLF-TSR cells. ChIP was performed with an increasing amount of anti-SAMHD1 antibody (Proteintech 12586-1-AP) or control IgG antibody (Santa Cruz sc-2027). ChIP-ed DNA was quantified on a dot-blot probed with a telomere specific probe and a control Alu-repeat probe. Dashed lines indicate a cropped image. (B) Immunoblot confirming pull-down of SAMHD1 during ChIP as shown in (A). (C) Co-IPs of SAMHD1 with TRF1 and TRF2 upon transient expression of the indicated 3xHA- and Flag-tagged constructs in HEK293T cells. Input fractions and anti-HA IPs were analyzed by immunoblotting for HA (BioLegend 901502) and Flag (Sigma F1804). The weak signal detected by anti-Flag antibody in the IPs without 3xHA-SAMHD1 likely represents background binding of TRF1/2 to the beads. Irrelevant lanes were omitted for clarity (dashed lines). (D) Co-IP of SAMHD1 with TRF1 as in (C). Where indicated, cell lysates were treated with ≥ 185 U/ml benzonase for 1 hour prior to IP.

If SAMHD1 depletion prevents telomere repair, what is the nature of the "outsider."

Reply: We now have determined that the outsider telomeres do not correspond to T-circles (Fig S9B). Thus, they are not generated due to inappropriate resolution of T-loops by nucleases as seen upon depletion of RTEL1 in telomerase positive cells (PMID: 29290468) or upon ectopic expression of TRF2ΔB (PMID: 29106411). As the outside telomeres stain positively for both G- rich and C- rich strands (see the image below), they appear to be double-stranded and thus do not correspond to single-stranded nascent DNA, displaced from stalled replication forks as observed by Coquel et al. in SAMHD1-depleted cells (DOI:10.1038/s41586-018-0050-1). Thus, the current analysis suggests that they are double stranded linear DNA fragments,

The vast majority of outside telomeres are positive for both C-rich and G-rich telomeric DNA. Representative metaphase chromosome spreads from HeLa cells depleted of TRF1 and SAMHD1 using shRNAs as in Fig 6. Telomeres are detected by FISH with FAM-[[TTAGGG]]₃ and Cy3-[[CCCTAA]]₃ probes, detecting the C-rich and the G-rich telomeric strand, respectively. DNA is stained with DAPI (blue). White arrows indicate outside telomeres (i.e. a telomeric signal positioned outside the DAPI signal).

The proposed model is reasonable; however, can the authors show that SAMHD1 interacts with the HR machinery or otherwise link it to the model?

Reply: see response to Ref. 1 - major concern #7.